# Role of HLA-I Structural Variants and the Polyreactive Antibodies They Generate in Immune Homeostasis

**DOI:** 10.3390/antib11030058

**Published:** 2022-09-08

**Authors:** Mepur H. Ravindranath, Fatiha El Hilali, Carly J. Amato-Menker, Hajar El Hilali, Senthamil R. Selvan, Edward J. Filippone

**Affiliations:** 1Department of Hematology and Oncology, Children’s Hospital, Los Angeles, CA 90027, USA; 2Emeritus Research Scientist, Terasaki Foundation Laboratory, Santa Monica, CA 90064, USA; 3Medico-Surgical, Biomedicine and Infectiology Research Laboratory, The Faculty of Medicine and Pharmacy of Laayoune & Agadir, Ibn Zohr University, Agadir 80000, Morocco; 4Department of Microbiology, Immunology, and Cell Biology, School of Medicine, West Virginia University, Morgantown, WV 26506, USA; 5Division of Immunology and Hematology Devices, OHT 7: Office of In Vitro Diagnostics, Office of Product Evaluation and Quality, Center for Devices and Radiological Health, Food and Drug Administration (FDA), Silver Spring, MD 20993, USA; 6Division of Nephrology, Department of Medicine, Sidney Kimmel Medical College, Thomas Jefferson University, Philadelphia, PA 19145, USA

**Keywords:** HLA-variants, epitopes, IVIg, monoclonal antibodies, polyreactive, immunostimulation

## Abstract

Cell-surface HLA-I molecules consisting of β2-microglobulin (β2m) associated heavy chains (HCs), referred to as Face-1, primarily present peptides to CD8+ T-cells. HCs consist of three α-domains, with selected amino acid sequences shared by all alleles of all six isoforms. The cell-surface HLA undergoes changes upon activation by pathological conditions with the expression of β2m-free HCs (Face-2) resulting in exposure of β2m-masked sequences shared by almost all alleles and the generation of HLA-polyreactive antibodies (Abs) against them. Face-2 may homodimerize or heterodimerize with the same (Face-3) or different alleles (Face-4) preventing exposure of shared epitopes. Non-allo immunized males naturally carry HLA-polyreactive Abs. The therapeutic intravenous immunoglobulin (IVIg) purified from plasma of thousands of donors contains HLA-polyreactive Abs, admixed with non-HLA Abs. Purified HLA-polyreactive monoclonal Abs (TFL-006/007) generated in mice after immunizing with Face-2 are documented to be immunoregulatory by suppressing or activating different human lymphocytes, much better than IVIg. Our objectives are (a) to elucidate the complexity of the HLA-I structural variants, and their Abs that bind to both shared and uncommon epitopes on different variants, and (b) to examine the roles of those Abs against HLA-variants in maintaining immune homeostasis. These may enable the development of personalized therapeutic strategies for various pathological conditions.

## 1. Introduction

Immune homeostasis is a delicately regulated balance of activation and suppression of immune and associated cells. Homeostatic imbalance may be both a cause and a consequence of pathological conditions ranging from inflammation, tissue injury, end stage organ disease, transplantation infection, malignancy to autoimmune diseases. In most immune conditions, there is no predictable pattern of recovery, and it may differ between individuals based on demographics as well as the lifestyle that include nutritional status, alcohol use, smoking, or recreational drug use. An in-depth immunological examination of the specific conditions may unravel the immune imbalance and enable developing personalized long or short duration therapeutic strategies to maintain immune homeostasis. 

Antibodies play a major role in immune homeostasis, depending on the nature of their target epitopes. An antibody’s hypervariable region in F(ab)2 is specific for the physico-chemical configuration of an epitope on an antigen. Such an epitope could be specific for one antigen or shared by several other antigens. One epitope being shared by hundreds of antigens is one of the unique characteristics of the HLAs. Therefore, the immune homeostasis is examined from the perspective of HLA variants and their corresponding antibodies (Abs). 

HLAs are highly polymorphic heterodimeric cell-surface molecules. Their polymorphism is exemplified by the six isoforms of HLA class-I (HLA-A, HLA-B, HLA-C, HLA-E, HLA-F, and HLA-G), and the three isoforms of HLA class-II (HLA-DR, HLA-DP and HLA-DQ), and by the thousands of identified alleles for each HLA-I and HLA-II gene and their respective proteins (Table 1). Several amino acid sequences of HLA polypeptide heavy chains (HCs) are highly specific for one allele or for one isoform, while some others are shared among other alleles of one isoform or all other isoforms of the HLA-class. In HLA-I, the shared epitopes are cryptic due to the presence of β2-microglobulin (β2m) and the tertiary and quaternary structures of HLA (HCs). Upon activation of immune cells under pathological conditions, the β2m-free HCs are generated on the cell-surface, which expose the cryptic shared epitopes. The exposed epitopes may elicit antibody response, in addition to binding with other ligands and Abs, depending on their antigenicity, immunogenicity, and other physicochemical characteristics.

Although it is known that HLA proteins “are centrally involved in the actions of the human immune system” [1], the Abs generated against allele-specific epitopes (monospecific Abs) among individuals may differ. However, the polyreactive Abs produced against shared epitopes of different alleles may be similar and remain as allo-HLA Abs in all humans. The strength and diversity of these polyreactive allo-HLA Abs may also vary depending on the epitopes they recognize.

A prerequisite to understanding HLA-mediated immunohomeostasis is to clarify the complexity of the HLA-structural variants under normal and pathological conditions and to distinguish the shared and uncommon epitopes recognized by the HLA-Abs. With this objective, this review examines: (i) the structural variations of HLA that make the shared epitopes cryptic or exposed under different situations depending on the pathophysiological conditions; (ii) the Abs generated against the shared and uncommon epitopes; (iii) the consequences of the interaction between the epitopes and their Ab; and (iv) the diversified immunoregulatory potential of the Abs. 

An understanding of the immunoregulatory roles of monospecific as well as polyreactive Abs is critical for developing personalized immunotherapies and vaccines to control various pathological conditions. The following categories are examined in this review:(i)Unique diversity of HLA polymorphism, which is reflected in the differences of the amino acid sequences in different domains, in the basic structural configurations, designated as HLA variants or Faces, and their role in immune homeostasis;(ii)Comparison of the diversity in allele specificity of HLA Abs in normal healthy individuals and in the commercial preparations of IVIg;(iii)Diversity of HLA monospecific and polyreactive monoclonal Abs (mAbs) generated by HLA molecules.(iv)Immune homeostatic role of HLA polyreactive Abs by binding
(a)to intact HLA (Face-1) (e.g., mAb W6/32); and(b)to the monomer HC of HLA (Face-2) (e.g., TFL-006 & TFL-007); and(v)Immune homeostatic role of HLA-monospecific Abs.

## 2. The Diversity of HLA Polymorphisms

### 2.1. Diversity in the HLA Molecules and the Domains of Their Polypeptides

An intact HLA-I molecule consists of a polymorphic αHC and a monomorphic β chain, non-covalently linked β2m. The β2m was first identified in the urine of patients with tubular proteinurea [2,3,4]. Then its molecular size and association with human HLA-I HCs were recognized [5,6]. The HLA-I αHC genes are located on the short arm of chromosome 6, whereas that of the monomorphic β2m is located on chromosome 15. In contrast, HLA-II molecules consist of two polymorphic HCs, α and β, differing in their amino acid sequences. HLA-I α-HC contains three domains, α1, α2 and α3, whereas HLA-II α and β HCs consist of four domains, α1, α2, and β1 and β2 (Figure 1).

### 2.2. The Domains Involved in the Primary Function of HLA

The primary function of the intact cell-surface HLA is antigen-presentation, and the domains serve as ligands for αβ T-lymphocyte receptors (TCRs). Most of the thymus-derived CD3+ T-cells express either the CD4 or the CD8 co-receptor molecules. CD4-expressing helper T-cells express TCRs specific for HLA class-II, whereas the CD8-expressing cytotoxic T-cells express TCRs specific for HLA class-I. The binding of the co-receptors of T-cells with HLA HCs enables the presentation of peptides to the major domain of the primary receptor of T-cells (Figure 1). The antigens presented by HLA-I are typically derived from intracellular infection caused by viruses. Presentation of these antigens to CD8+ T-cells initiates the cytolytic killing of the infected cells to prevent viral replication and spread. Since all human cells are potential targets for viral infections, the HLA-I isomers are expressed on all types of nucleated cells.

The antigens presented by HLA-II are derived from pathogens present in the extracellular spaces, which include bacteria, viruses and other pathogens. This presentation stimulates CD4+ cells to activate other immune cells, such as macrophages and B-cells. Hence, HLA-II is selectively expressed on professional antigen-presenting cells, which include macrophages, dendritic cells, and B cells. HLA-II antigen presentation activates CD4+ T-cells that stimulate the macrophages to release cytokines to further activate macrophages directly and help B cells to produce opsonizing Abs.

Essentially, the CD4 receptor binds to the β2 domain of the βHC. CD8-receptors of cytotoxic T-cells bind to α3 of the αHC (Figure 1). The amino acid sequences of α3 among all HLA-I HCs of isomers are almost identical with few variations. Similarly, the shared amino acid sequences of the β2-domain among all HLA-II HCs of isomers are strikingly similar. Table 2 illustrates the shared sequences in the α3-domain of HLA-I and the β2-domain of HLA-II that bind to T cell co-receptors CD8 and CD4, respectively. Figure 2 and Figure 3 show the sequences of α3 domain (in red) and also illustrate basic similarities and dissimilarities in the amino acid sequences of classical HLA-I (HLA-Ia: HLA-A, HLA-B and HLA-C, and non-classical HLA-I (HLA-Ib: HLA-E, HLA-F and HLA-G). The amino acid sequences in the α1 and α2 domains of HLA-I, and the α1 and β1 domains of HLA-II show noteworthy variability among different alleles. Such variations enable their recognition of specific peptides for antigen presentation. Amino acid sequences (shown in green) refer to the α1 domain of HLA-I, the region involved in formation of the groove and binding site of peptides. The differences seen in the specific amino acids in the sequence of the α1 domain are highly variable among different alleles of HLA-A, B, C, E, F and G (Figure 4A–C). Similarly, the sequences in the α2 domain of HLA-A, B, C, E, F and G (Figure 4D–F) are unique in that they contain a region involved in the formation of the grove for binding site of peptides and a long sequence (cryptic) domain masked by β2m (Figure 5). 

The amino acid sequences of α2-domain that are hidden by β2m (Figure 5) constitute unique cryptic epitopes, and, most importantly, they contain sequences (such as ^141^AYDGKDY^147^ and ^152^EDLARSWTA^159^) (shown in black in Figure 2) shared by almost all HLA-I isomers. Strikingly, the amino acid sequences are identical in all alleles of all six HLA-I loci (Figure 4D–F). Whereas these sequences are hidden from exposure (cryptic) by β2m in intact HLA-I (also known as Closed conformers or Face-1), they remain exposed in naturally occurring β2m-free α-HCs (also known as Open conformers or Face-2 HLA). Interestingly, the antigenicity, flexibility, hydrophobicity, and beta turn characteristics are highest for the AYDGKDY sequence compared to any other shared sequences (Table 3). The exposure of cryptic epitope in Face-2 is overexpressed in activated immune cells, resulting in immune recognition. Similarly, the sequence ^161^DTAAQI^166^ present in all alleles of HLA-B, HLA-C, HLA-E, and HLA-G, differs in position 162 in HLA-A as ^161^DMAAQI^166^ (Figure 4D) and in position 163 in HLA-F as ^161^DTVAQI^166^ (Figure 4F). 

Note especially the peptide sequence AYDGKDY, which is cryptic and shared by all isoforms of HLA-I. The bioinformatics analysis was carried out by using the Immune Epitope Database (IEDB) to predict antigenicity rank of epitopes. Chou and Fasman beta turn, Kolaskar and Tongaonkar antigenicity, Karplus and Schulz flexibility and Parker hydrophilicity prediction methods in IEDB were employed. The methods predict the probability of specific sequences in HLA-E that bind to Abs being in a beta turn region, being antigenic, being flexible, and being in a hydrophilic region. Antigenicity rank is calculated by pooling the probability values.

### 2.3. The Four Structural Variants of HLA-I

In addition to naturally occurring Face-1 and Face-2, there are two additional variants formed naturally from the Open conformers and all these variants are referred to as the four faces of HLA-I [11], as illustrated in Figure 6. Elucidation of these four faces of HLA-I variants is necessary to comprehend HLA-mediated immune homeostasis. These faces are as follows:Face-1: The intact HLA-I that consists of an α-HC non-covalently linked to β2m;Face-2: HLA-I monomeric α-HC devoid of β2m, exposing immunogenic cryptic sequences, such as ^141^AYDGKDY^147^, ^152^EDLARSWTA^159^, and ^161^DTAAQI^166^.Face 3: Homodimers of Face-2: Dimers of two α-HCs of the same allele formed by S-S linkage of cysteines at position 67, as observed in HLA-B27.Face-4: Heterodimers of Face-2: Two α-HCs of different HLA-I monomers formed by linkages of cysteines at positions 67, 101, 164, 203, or 43, as in HLA-G, or by salt linkage, H-bonding, or van der Waal forces, as observed in HLA-F and HLA-C.

**Figure 6 antibodies-11-00058-f006:**
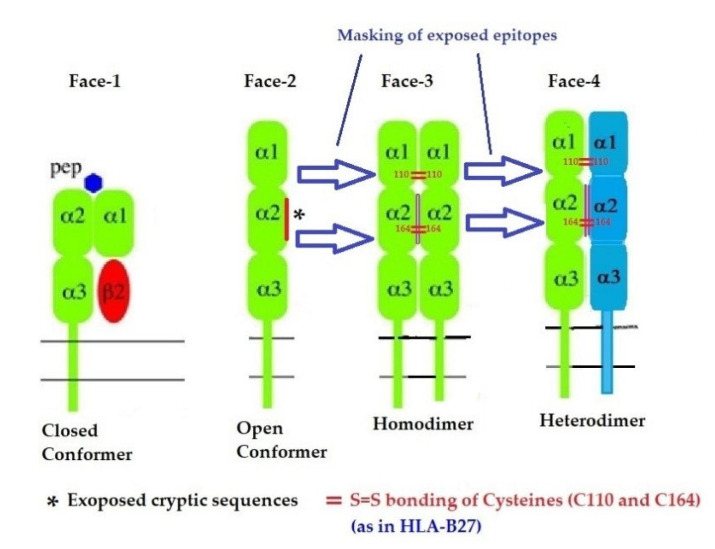
Four major structural configurations or faces of HLA-I. The shared epitopes (see Table 3) remain masked by β2-m in Face-1 or intact HLA. In the absence of β2m, the HLA becomes monomeric (Face-2) on activated immune cells and on cancer cells, resulting in the exposure of cryptic sequences. It appears to be for a shorter duration since the exposed sequence is masked by homodimerization or heterodimerization. During dimerization, there is an increased propensity for pairing of cysteines and formation of disulfide bonds (as in HLA-B27, HLA-C and HLA-F), a mechanism to remask the exposed sequences.

### 2.4. Transitory Expression of Face-2 on Activated Immune Cells 

The presence of naturally occurring β2m-free HCs of HLA-I, Face-2, was first recognized in the human B lymphoblastoid cell line T5-1 [13]. Cell lysate was immunoprecipitated with an anti-H polyclonal serum, which does not bind to Face-1, but only to β2m-free HCs of HLA-I (Face-2). Only 1 to 2% of the HCs (Face-2) present on the surface of T5-1 cells were precipitable by anti-H. Since this investigation, cell-surface expression of Face-2 has been observed across lymphocyte subpopulations (CD3+ T-cells, CD19+ B-cells, CD56+ NK cells and CD14+ monocytes), and importantly these HCs are particularly overexpressed on activated cells, as well as on extravillous trophoblast, T-cells and monocytes [14]. Face-2 was also observed on the cell-surface of both in vitro and in vivo activated human T-cells, but not on the resting ones [12]. Immunoprecipitating the activated T-cells with both the mAb W6/32 and the mAb LA45 confirmed that W6/32 precipitated Face-1, whereas LA45 precipitated Face-2 only. Western blot analysis of the cell lysates further confirmed these results [12]. Face-2 is also expressed on human peripheral blood T-cells activated either by phorbol myristate acetate (PMA), anti-CD3 antibody, PHA, brefeldin A (PFA), or IFN-γ [15]. Furthermore, using purified and biotinylated mAb L31, HLA-C Face-2 was recognized on activated T-cells [16]. The possibility of formation of Face-3 by homodimerization of 2 identical Face-2 molecules was dependent on the unpaired cysteine at position 67 (Cys67) [17,18]. Homodimerization of Face-2 of HLA-B27 was recognized using mAbs HC-10 and ME1 [19].

The formation of Face-4 by heterodimerization of two different Face-2 molecules was reported in HLA-F [20]. Any HLA-I HC could interact with HLA-F only when the latter was in the form of Face-2. This interaction was directly observed by coimmunoprecipitation and by surface plasmon resonance and indirectly on the surface of cells through tetramers and HLA-I Face-2 colocalization. The data indicate that HLA-F is expressed as Face-2, and it has physical interaction with HCs (Face-2) of the same allele of HLA-F (Face-3) and with other alleles and isoforms of HLA-I (Face-4), based on the differential coimmunoprecipitation experiments. At least three different forms (Face-2, Face-3, Face-4) of HLA-F based on differential staining of surface HLA-F using mAbs 4A11 and 3D11 (HLA-F HC specific) and 4B4 (specific for intracellular HLA-F) over the course of lymphocyte activation were observed [20]. It is speculated that “HLA-F and MHC-I HC interactions can occur *in trans* between cells” (16, p. 6206). HLA-F expressed as Face-2 is not recognized by the mAb W6/32; however, the homodimer (Face-3) of HLA-F was recognized on the cell-surface by W6/32, particularly in the absence of β2m. Furthermore, it is reasonably well established that Face-2 is overexpressed in human cancers such as neuroblastoma, renal cell, colon, gastric, breast, ovarian, bladder carcinoma and melanoma, using mAbs L31, LA-45, LHC10 and M38 [21,22,23,24,25,26]. These reports led to the hypothesis that homo- and heterodimers of Face-2 exist in human cancers, and this deserves serious attention in the potential development of autologous and allogeneic tumor cell vaccines with various adjuvants.

### 2.5. Do the Structural Variants of HLA Have a Role in Immune Homeostasis?

Face-2 exposes cryptic epitopes and leads to immunogenicity (recognition by B cells and eliciting antibody response) and antigenicity (recognition by Abs formed against the cryptic epitopes). Both homodimerization and heterodimerization of Face-2 facilitates masking of the exposed (previously cryptic) epitopes of Face-2, such that they are blocked from immune recognition by the “polyreactive Abs”. Evidently, Face-2 seems to be a short-lived structural variant of HLA. In order to unravel the roles of the structural variants of HLA-I and the generated polyreactive antibodies in immune homeostasis, the following questions are highly pertinent:I.Is the dimerization of Face-2 a natural mechanism to prevent the exposure of cryptic epitopes otherwise masked by β2m?II.If so, what is the purpose of ephemeral exposure of these cryptic epitopes?III.Does the transient-exposure of cryptic epitopes of Face-2 generate polyreactive Abs following immune recognition by B cells? If so, what kinds of polyreactive HLA Abs are being produced?IV.Do the naturally occurring polyreactive Abs bind to the ephemeral exposure of cryptic epitopes on Face-2 to elicit salient immunoregulatory functions?V.How do the immunoregulatory effects differ between the binding of Abs directed against Face-1, such as that of mAb W6/32, and that of Abs directed against Face-2, such as that of TFL-006 and TFL-007?

Answering these questions may unravel the roles of the structural variants of HLA-I and the polyreactive Abs they generate in immune homeostasis.

## 3. The Diversity of HLA-I Antibodies

### 3.1. The Diversity of Natural Anti-HLA-I Abs in the Normal Human Sera

Avrameas et al. [27] observed Abs directed against several autoantigens such as tubulin, actin, thyroglobulin, myoglobulin, fetuin, transferrin and albumin and hypothesized that naturally occurring IgG-Abs “against a high variety of self antigens” may occur in normal healthy individuals. Indeed, Collins et al. [28] first reported a naturally occurring anti-HLA-A8 Ab. Tongio et al. [29] confirmed the presence of natural IgM Abs in the sera of 21 normal healthy individuals reacting with HLA-I alleles. Subsequently, more reports [30,31] emanated on naturally occurring Abs to HLA-I and HLA-II. Since these Abs in the sera of normal and healthy individuals recognize several HLA alleles not found in oneself, they were considered as allo-HLA Abs. Morales-Buenrostro et al. [32] examined in detail these anti-HLA-I Abs in the sera of 424 nonalloimmunized healthy males using a single antigen bead assay (SAB assay) with Luminex beadsets coated with more than 90 different purified recombinant HLA molecules. They found allo-HLA Abs to class-I in 42%, class-II in 11%, and both in 12 of the individuals examined. Interestingly, it was reported that the serum Abs also bound to beadsets treated with a mild acid to convert Face-1 to Face-2, suggesting that the normal non-alloimmunized male sera contain Abs reacting to both Face-1 and Face-2. This is the first report documenting the occurrence of Abs against both Face-1 and Face-2.

We have examined the sera of normal healthy males (non-alloimmunized) and females for the presence of IgM and IgG reacting to HLA-A, HLA-B, HLA-C, HLA-E, HLA-F, and HLA-G using the SAB assay on the Luminex platform on single antigen beads (SABs) provided by different vendors [33,34]. Our focus was to distinguish between serum HLA Abs reacting to Face-1 and Face-2. The beadsets differed significantly in the composition of the antigens present on their surface. LIFECODES SABs manufactured by Immucor (Norcross, GA, USA) contain only intact HLA (Face-1) [33], whereas the LABScreen SABs (Thermofisher-One Lambda Inc., Canoga Park, CA, USA) contain both Face-1 and Face-2 [34].

We have examined both IgM and IgG in the sera of normal healthy males (non-alloimmunized) and females. Table 4 documents the profile and strength (expressed as mean fluorescence intensity or MFI) of HLA-A, HLA-B and HLA-C reactivities of the normal non-alloimmunized male sera. Since IgG in human sera can occur as immune complexes in conjunction with anti-idiotypic IgG or IgM Abs and other proteins, we have also examined the normal sera after purifying the Abs with a Protein-G affinity column. Table 5 documents the profile of serum anti-HLA reactivity after removing the immune complexes and other factors bound to IgG. It is interesting to note that the IgG Abs in these sera bound to almost all alleles of HLA-A, HLA-B and HLA-C. We have also examined the sera of several normal healthy male and female volunteers for IgM and IgG reactivity to HLA-E, HLA-F and HLA-G. Figure 7A–C illustrate the relative proportions (MFIs) of IgM and IgG reacting to HLA-E, HLA-F, and HLA-G in the unpurified sera of the normal individuals. The relative ratios of anti-HLA-Ib IgM and IgG in the sera of normal males (non-alloimmunized) and females are depicted in Figure 8. Note the high prevalence of IgM-Abs against HLA-E in a number of males and females, and the high prevalence of IgG reacting to HLA-G. 

### 3.2. The Diversity of Abs in Intravenous Immunoglobulin

The presence of natural allo-HLA Abs in human plasma implies that intravenous immunoglobulin (IVIg), the IgG purified and pooled from the plasma of thousands of normal donors, contains such natural HLA antibodies. IVIg has been extensively used to prevent microbial and fungal infections and to reduce inflammation after transplantation [35,36,37,38,39] and in other inflammatory and autoimmune diseases [40,41,42,43]. IVIg is often administered to lower the level of anti-HLA Abs in end-stage organ disease patients and transplant recipients [40,41,42,43,44,45,46,47,48,49,50,51,52,53,54]. These findings lead Kaveri and his team [55] to propose a role for IVIg in immune homeostasis. The different vendors’ commercial preparations of IVIg are known to suppress activated CD4+ T-cells and their production of proinflammatory cytokines when the cells are activated by various stimuli in vitro [56,57,58,59,60,61,62,63,64,65,66]. Interestingly, Kaveri et al. [67] reported that IVIg has Abs, to a conserved region of HLA-I, that are capable of modulating CD8+ T cell mediated function. We have examined the presence of Abs against different alleles of HLA-I in different therapeutic preparations of IVIg [68] and tested IVIg HLA-I reactivity on a Luminex platform using three different beadsets: (1) Regular LABScreen SABs, which are coated with Face-1 admixed with Face-2; (2) the mild acid-treated Regular LABScreen SABs exposing the cryptic epitopes and only carry Face-2; and (3) iBeads, which are mildly trypsinized regular LABScreen SABs for removal of Face-2 so that they carry only Face-1. The results (Figure 9) confirm that IVIg reacts not only with Face-1 but also to Face-2. However, all preparations of IVIg showed greatly reduced reactivity for iBeads, suggesting that they predominantly recognize the Face-1 of HLA-I. At this stage, it is still uncertain whether the immunoregulatory properties of IVIg are due to the Abs binding to Face-1 or Face-2 of HLA.

It was necessary to delineate the functions of HLA Abs by generating HLA polyreactive monoclonal Abs mimicking the Face-2 polyreactive IVIg. As was reported for IVIg, there were infrequent reports indicating that anti-HLA mAbs suppressed T cell proliferation [69,70,71,72], T cell activation [69], interleukin (IL)-2 and IL-2R synthesis [70], and apoptosis induction [73,74]. These reports, except for the mAb W6/32, did not identify the specific epitopes or amino acid sequences recognized by the anti-HLA-I mAbs. 

Not only were we able to develop mAbs mimicking HLA-I reactivities of IVIg, but also documented that these IVIg-mimetics perform the unique immunoregulatory roles of IVIg similar to or better than the current formulation of IVIg while minimizing the adverse effects of IVIg. Possibly, they may minimize the adverse effects of IVIg by eliminating other non-HLA Abs. 

## 4. The Pattern of Diversity of HLA Monoclonal Abs (mAbs)

### 4.1. Is the mAb W6/32 Specific for Closed Conformers (Face-1) of HLA-I?

Many investigators studying the functional role of HLA-I Abs [75,76,77,78,79,80,81,82,83,84,85,86,87,88,89] have used a well-known murine monoclonal antibody W6/32 on both naïve and activated cells that express Face-1. The mAb W6/32 has been used for quality control by the manufacturers of HLA beadsets (One Lambda Inc, Thermo Fisher, and Immucor) to prove that Face-1 are coated on the beadsets. The mAb W6/32 (IgG2a) has been developed to monitor HLA-I on the human cell-surface, since the mAb reacted with a wide range of cells but not with Daudi Burkitt lymphoma cells, which are devoid of β2m-associated HCs [75,76,77,78,79,80,81,82,83,84,85,86,87,88,89,90,91,92,93]. The same team established that W6/32 immunoprecipitated both a 43kDA HLA HC and a 12-kDa β2m. Interestingly, it formed immune complexes with β2m-associated HCs of HLA but not with β2m-free HCs. The mAb bound to both peptide-carrying β2m-associated HCs and peptide-free β2m-associated HCs [94,95]. However, W6/32 failed to bind to HC-associated β2m in the MHC (Major Histocompatibility Complex) class-I of mice, rats, rabbits, and guinea pigs [96,97,98,99,100]. Interestingly, when these non-human MHC HCs were reconstituted with β2m of cattle or humans, they regained the binding affinity. A comparative study of the amino acid sequences of the HCs and β2m of all animal MHC antigens with human HLAs identified the critical epitope as amino acid residue 121 in the HCs of HLA-I, the glutamic acid and glutamine residues in β2m at positions 44 and 89, respectively, [101] and the amino acid residue at position 3 [102].

Most interestingly, Tran et al. [103] reported that the mAb W6/32 “actually recognizes an epitope present on isolated non-reduced α-chains of most HLA-B allelic forms.” The specificity of W6/32 for Face-2 of HLA-B raises concern about the use of mAb W6/32 to validate the expression of intact HLA-B coated on the LABScreen and Immucor beadsets used on the Luminex platform. 

### 4.2. Are There True HLA-I Polyreactive mAbs That Recognize Face-2 of All HLA-I Isoforms and Their Alleles?

To develop mAbs for HLA-E for immunodiagnosis, we have used recombinant β2m-free HCs (Face-2) of HLA-E^R107^and HLA-E^G107^ (Immune Monitoring Laboratory, Fred Hutchinson Cancer Research Center) [104]. The mAbs generated were analyzed using multiplex bead assays on a Luminex platform for HLA-I reactivity. Beads coated with an array of HLA heterodimers admixed with HCs (LABScreen) were used to examine the binding of IgG to different HLA-Ia (31 HLA-A, 50 HLA-B, and 16 HLA-C) and HLA-Ib (2 HLA-E, one of each HLA-F and HLA-G) alleles, as detailed elsewhere. A striking diversity in the HLA-Ia and/or HLA-Ib reactivity of mAbs was observed. The hybridomas and their mAbs obtained are shown in Table 6.

We have reported a detailed analysis of HLA-Ia and -Ib reactivities of a monospecific mAb (TFL-033) [104,105,106,107,108,109] and the polyreactive mAbs (TFL-006 and TFL-007) [109,110,111,112,113]. The binding affinity of the monospecific mAb TFL-033 was tested using the HLA-E specific amino acid sequence (SARDATA) as well as the binding affinity of the mAb TFL-006 with the most shared sequence ^141^AYDGKDY^147^, by inhibiting the binding of mAbs to the beadsets (LABScreen) used on Luminex Platform. The affinity of the polyreactive mAb TFL-006 to shared sequences which are cryptic in Face-1 but exposed in Face-2 was tested. For this purpose we have compared the mAb-binding to LABScreen (LS) beadsets which contain Face-1 admixed with Face-2, to acid treated LS SABs which contain only Face-2, and to iBeads [33] or to LIFECODES (LC) [34] SABs, which contain only Face-1. The results are recapitulated in Table 7. The mAb TFL-006, which binds to exposed cryptic epitopes found in Face-2, did not recognize theFace-1 HLA-I molecules coated on the iBeads or LC SABs. Interestingly, the mAb TFL-006 recognized regular LS SAB lots, confirming the presence of Face-2 on the regular LS SABs. The results revealed explicitly that the mAb TFL-006 only recognize the shared peptides when exposed on Face-2 HLA, but not when cryptic on the Face-1. 

## 5. The Immunoregulatory Potential of Anti-HLA-I mAbs unravels the Roles of the HLA-I Face-1 and Face-2 and Their Polyreactive Abs in Immune Homeostasis

### 5.1. The Immunopotential of Anti-HLA Face-1 mAb W6/32

Several mAbs were used on the assumption that they are directed against intact HLA-I (Face-1). Most of them are poorly defined and used by few groups, without characterizing their epitope specificity. The most common mAb used to localize intact HLA-I (Face-2) on the cell-surface is mAb W6/32, which binds to both the HCs and β2m of all HLA-I isomers. Interpreting the affinity of the mAb requires caution in view of two findings:(i)Reports published from 1989 [12,15,16,17,18,19,20,24,25,26,114,115,116,117,118,119,120,121] document the prevalence of Face-2 on activated immune cells. Publications on W6/32 have ignored these findings until 2001.(ii)Tran et al. [103] reported that “although it (W6/32) does recognize in a monomorphic fashion all HLA-A, B, and C molecules when present in their native state (Face-1) on the cell-surface, under partially denaturalized conditions, it recognizes an epitope preserved in free nonreduced α chains of most HLA-B antigens but not in other HLA class-I α chains. Essentially identical results were also obtained with another pan-HLA class-I mAb MEM-147 which cross-blocks W6/32” (p. 443, column 1, para 3). Therefore, one should be cautious while interpreting the results stemming from using W6/32 on activated cells, which express Face-2.

Two categories of studies shed light on the immunoregulatory role of the mAb. The first one investigates the impact of W6/32 on endothelial cells (ECs), smooth muscle cells (SMCs), and epithelial cells. The second one investigates the effects of the mAb made on immune cells, particularly CD3+/CD4+ and CD3+/CD8+ T lymphocytes and CD3+/CD19/20+ B lymphocytes.

#### 5.1.1. Impact of the mAb W6/32 on Endothelial, Smooth Muscle and Epithelial Cells 

Augmentation of serum anti-HLA IgG during Transplant Atherosclerosis (TA) results as a consequence of the intimal proliferation of smooth muscle cells, endothelial cells, and fibroblasts, leading to vessel obstruction, fibrosis, and graft loss. Reed and her team [76,77,78,79,80,81,82,83,84,85,86,87,88,89] investigated the role of the anti-HLA-I Abs in TA. On the assumption that the mAb W6/32, represents a typical anti-HLA-I IgG Ab, they studied its impact on the activation and proliferation of endothelial cells (ECs) and smooth muscle cells (SMCs). Indeed, the binding of W6/32 to ECs increased tyrosine phosphorylation of intracellular proteins, inositol phosphate generation, and cell proliferation [77,78]. In addition, W6/32 added to ECs and smooth muscle cells (SMCs) in vitro increased mRNA expression of high affinity fibroblast growth factor receptor (FGFR-1), and enhanced basic fibroblast growth factor ligand binding, culminating in augmented cell proliferation [76,78]. The FGFR-1 is a member of the superfamily of tyrosine kinase growth factor receptors and is capable of transducing proliferative signals. These findings suggest that the binding of W6/32 to cell-surface HLA-I is able to transduce signals and generate intracellular messengers. Furthermore, the treatment of ECs with mAb W6/32 stimulated a higher level of FGFR expression than treatment with serum Abs to individual A locus or B locus molecules, suggesting that the intensity of the signal transduction is related to the aggregation of HLA-I molecules on the cell-surface [76,78]. Evidently, the mAb W6/32 promotes cell proliferation and upregulates cell survival genes. The intracellular events initiated by W6/32 binding are concentration-dependent. At high concentration, the mAb upregulated cell proliferation, whereas at low levels, the mAb activated the PI3K/Akt pathway and promoted expression of cell survival proteins including Bcl-2 and Bcl-xL. These findings were further substantiated by others [122]. 

Similarly, Mohanakumar’s group [123,124] reported that the incubation of airway epithelial cells (AECs) with the W6/32 induced a significant cell proliferation within 24 h comparable with the proliferation observed in the presence of LHC-9 growth medium. However, the proliferation of the AECs induced by the W6/32 was only observed within the first 24 h. AECs treated with W6/32 exhibited significantly greater proliferation and tyrosine phosphorylation of proteins compared to AECs treated with normal human serum or mouse IgG, respectively.

#### 5.1.2. Impact of the mAb W6/32 on Immune Cells 

Ferrone and his team [125,126] observed that W6/32 enhanced the T cell proliferation induced by CD2-specific mAb 9.1 in a dose-dependent manner, which was not abrogated by the addition of exogenous IL-1 and IL-2. However, no such detectable enhancement was seen with CD2-induced stimulation by anti-β2m mAb NAMB-1. Ferrone’s team [127,128] also observed that W6/32 enhanced the proliferation of CD1- CD3+ CD4+ CD8+ T-prolymphocytic leukemic cells induced by CD2-specific mAb 9.1. However, W6/32 inhibited the proliferation induced by the CD3 binding mAb OKT3. These findings are supported by others [129,130,131]. Figure 10 summarizes the role of mAb W6/32 in enhancing the CD2- and CD3-mediated activation of T-cells. 

It is known that HLA-I regulates T cell proliferation induced via both CD2 and CD3 pathways [132]. Evidently, the mAb W6/32 may ligate with both Face-1 and Face-2 of T-cells activated via CD2 and CD3 molecules on the T-cells. When the T-cells are activated by the ligation of the mAb 9.1 on CD2, the binding of W6/32 enhances proliferation. However, when the T-cells are activated via CD3 by the ligation of the mAb OKT3, the proliferation is suppressed. These findings suggest that the mAb W6/32 may influence HLA-I on the cell-surface differently, one upregulating intact HLA (Face-1) and another downregulating Face-2. Accordingly, W6/32 may enhance the proliferation of CD2-specific mAb-mediated activation of T-cells and suppress proliferation of CD3-mediated activation of T-cells. Examining HCs-polyreactive mAbs binding specifically to Face-2 may provide a better understanding of this aspect.

The findings of Tran et al. [103] clarified that β2m is not a prerequisite for W6/32 binding to HLA. However, it is still not clarified why that is applicable to β2m-free HCs of HLA-B and not applicable to β2m-free HCs of HLA-A or HLA-C. Since HLA-E, HLA-F and HLA-G were not tested, there is a long way to go to fully define the specificity of mAb W6/32. 

### 5.2. The Immune-Reactive Potential of HLA-I Shared-Epitope Polyreactive mAbs (TFL-006/TFL-007) Directed against Anti-HLA Open Conformers (Face-2) 

T-cells can also be activated by phytohemagglutinin (PHA), phorbol myristate acetate (PMA), anti-CD3 antibody, brefeldin A (PFA), and IFN-γ. Both CD2 and CD3 domains on T-cells may serve as receptors for several ligands such as PHA and get activated, resulting in proliferation. Such activated T-cells overexpress β2m-free HCs (Face-2) [12,15,16,17,18,19,20,24,25,26,55,56,57,58,92,93,95,96]. Upregulation of Face-2 upon activation of T-cells could also be related to CD3-mediated activation.

In naturally occurring β2m-free HCs (Face-2), cryptic epitopes hidden by β2m in Face-1 are exposed primarily in the α2 domain. This domain (Figure 2, Figure 3, Figure 4D–F Figure 5) contains several sequences (such as ^141^AYDGKDY^147^) shared by HLA-A, -B, -C, -E, -F and -G. (Table 3). The shared epitopes of all HLA isomers can be better appreciated when fitted on the same page, as shown in Figure 4.

Given the nature of the shared epitopes, mice were immunized with recombinant HCs of HLA-E and the Abs elicited after immunization were examined for binding to different alleles of different HLA isomers [104,105,106,107,108,109]. A profile of the antibody specificities of more than 200 mAbs are illustrated in the Table 6 and are described in detail elsewhere [104,105,106,107,108,109]. Distinctly, six mAbs were highly polyreactive to β2m-free HCs (Face-2) of all alleles of HLA isoforms. The shared epitopes were identified, and the synthetic amino acid sequences were used to inhibit the binding of the polyreactive mAbs [109,110,111,112,113]. Of these, TFL-006 ranks first and TFL-007 ranks second in their strength of polyreactivity as illustrated [68,110,112,113]. We will focus on their immunoreactive potential and compare it with (1) mAb W6/32 and (2) IVIg.

#### 5.2.1. HLA-I Face-2 Polyreactive mAbs (TFL-006/TFL-007) Mimic IVIg HLA-I Reactivity and Regulate Activated Lymphocytes

Immunologically relevant shared epitopes are located in the α2-domain of the heavy chain of HLA-I, as illustrated in Figure 2, Figure 3, Figure 4D,E, Figure 5 and Figure 6. These sequences are cryptic in Face-1 but exposed in Face-2 (Figure 5 and Figure 6). To determine how the recognition by mAbs of the shared epitopes of Face-2 impacts proliferation of activated cells, we have added mAbs TFL-006 and TFL-007 to human T-cells activated by PHA-P (PHA-*Phaseolus*) in vitro [133]. Two sets of experiments were done in triplicate on both CD4+ T-cells and CD8+ T-cells as follows:
(i)The impact of the mAbs (TFL-006 and TFL-007) (Experimental) and controls (TFL-033 and TFL-037) (Controls) on the blastogenesis in PHA-treated and untreated T-cells was determined by counting lymphoblasts after culturing purified lymphocytes from donors for 72 h. Lymphoblasts were recognized by flow cytometry based on size (side-scatter) and granularity (forward scatter). The blastogenesis and suppression were compared with different commercial preparations of IVIg. (ii)The impact of the above listed mAbs was examined for enhancement or suppression of proliferation by labelling the purified lymphocytes during PHA activation with the intracellular fluorescent dye carboxyfluorescein succinimidyl ester (CFSE). Using flow cytometry, the mitotic activity was measured by the successive twofold reductions in fluorescent intensity of the T-cells placed in culture for 72 h. PHA-treated T-cells undergo, on average, four to six divisions by 72 h. The suppression of blastogenesis and proliferation can be visualized by cessation of progression of mitotic activity, as measured by the successive twofold reductions in the fluorescent intensity after 72 h of treatment.

The findings are elaborated in detail in previous reports [111,112,113]. Figure 11 and Figure 12, and Table 8 elaborate the results of dosimetric suppression of blastogenesis and proliferation of PHA-activation of both CD4+ and CD8+ T lymphocytes by anti-HLA-E mAbs (TFL-006 and TFL-007) that mimic the HLA class-Ia and-Ib reactivity of IVIg. The IVIg-mediated dosimetric suppression of blastogenesis and proliferation of PHA-activated T-cells was identical to that of the HLA-polyreactive mAbs, whereas the control anti-HLA-E mAbs (TFL-033 and TFL-037) did not have any impact on PHA-activated T-cells. Interestingly, the mAbs TFL-006 and TFL-007 are shown to be better suppressors of blastogenesis of the PHA-activated CD4+ T-cells than IVIg. The concentration of anti-HLA-E mAbs required for this suppression was as much as 150-fold lower than that of IVIg.

HLA-I open conformer (Face-2) polyreactive monoclonal Abs (mAbs). The mAbs were generated following immunization with recombinant HCs of HLA-E^R^. The density of the population of CD4+ T cell (**A**) and CD8+ T cell (**B**) cultured with and without PHA, and after adding TFL-037s (control mAb) at 1/10 dilution, TFL-006s (1/10) and TFL-007s (1/10 and 1/100) to the PHA-treated CD4+ T-cells. The values are expressed as the mean of triplicate analyses +/− standard deviation (s.d.) and two-tailed *p*-values. The two-tailed *p*-values, if significant, are indicated by a horizontal line connecting the two groups. Note that the PHA-activated CD4+ T lymphocytes (**A**) remained unaffected by treatment with mAb TFL-037s, and that only lymphoblasts (not resting or naïve T-cells) decreased after treatment with mAb TFL-006s. Similarly, the lymphoblasts (not resting or naive T-cells) decreased significantly after treatment with mAb TFL-007s at 1/10, and after treatment with mAb TFL-007s at 1/100 dilution. Clearly, both anti-HLA-E mAbs TFL-006s and TFL-007s significantly suppress PHA-activated CD4+ T lymphoblasts, although the dosimetric suppression by TFL-007s is more striking. Similarly, the lymphoblasts of CD8+ Naïve T-cells (**B**) decreased after treatment with both mAbs but the decrease is significant with mAb TFL-007. Interestingly, mAb TFL-007 also suppressed PHA-activated naïve T-cells significantly. Note: “s” after mAb numericals (e.g., TFL-006s) refers to culture supernatant of the mAb.

Figure 13 illustrates the possible mechanism of suppression of PHA activated T-cells mediated by mAbs directed against shared epitopes exposed in Face-2. The proposed mechanism is based on the Mustelin model for T-cell activation [134] and can be explained by the molecules shown on the cell surface. The CD3, CD4, IL-Rα and T cell receptor molecules are expressed on the cell surface. PHA interacts with surface ligands on CD3 and initiates phosphorylation of the cytoplasmic domain of CD3 leading to activation of transcription factors. Such a cascade of events results in the production of cell surface molecules such as IL-2Rα [135,136] and Face-2 [12,15,16,17]. Face-2 on the cell surface exposes the cryptic epitopes. The binding of HLA-I polyreactive Abs (e.g., mAbs TFL-006 & TFL-007) to the shared epitopes promotes the phosphorylation of Tyr 320 [137,138,139] and Ser 355 [140] in the cytoplasmic domain of Face-2. The phosphorylation mediated by the HLA-I polyreactive Abs leads to activation of phosphatases leading deactivation of transcription factors and abrogation of the synthesis of proteins involved in blastogenesis and mitosis [141,142,143,144], as portrayed in the Figure 13. 

#### 5.2.2. HLA-I Face-2-Polyreactive mAbs (TFL-006/TFL-007) May Simulate IVIg in Suppressing allo-HLA Abs Produced by Human B Cells

IVIg, introduced as replacement therapy for patients with primary and secondary immune deficiencies [55,68], was extensively used for treatment of several autoimmune and systemic inflammatory diseases, acute and chronic relapsing diseases, and autoimmune disorders mediated by autoaggressive T-cells [35,36,37,38,39,40,41,42,43]. Additionally, IVIg modulates B-lineage cell antibody production. The use of IVIg for allograft recipients was approved by the US Food and Drug Administration (FDA) to reduce anti-HLA antibody levels prior to transplantation and to reverse humoral rejection [44,45,46,47,48,49,50,51,52,53,54]. 

All commercial IVIg preparations are reactive to almost all HLA-I antigens [55,145] (Figure 9), mimicking the HLA-I polyreactive mAbs (Table 7). Simulation of the antibody suppressive activity of IVIg by the polyreactive HLA mAbs was tested by comparing the efficacy of IVIg versus that of the mAb TFL-007 in suppressing the allo-HLA Abs production by hydridoma cell line (HML16), developed from a postpartum alloimmunized woman. The HML16 cell line produced anti-HLA class-I alloAbs with differing MFIs: high against B*07:02, B*81:01, B*67:01 and B*42:01; and low against B*27:08, B*27:05, B*55:01, B*56:01 and B*82:01.

The effects of two different therapeutic IVIg preparations (GamaSTAN and Gamunex) and the HLA-polyreactive mAb TFL-007 were tested on HML16 at different dilutions and protein concentrations and compared with the control media. Neither of the two IVIg preparations showed any significant suppression of the allo-HLA-B IgG production by HML16 cells. In contrast, the mAb TFL-007 showed a dose-dependent, (at 1:10 (100 μg/mL), 1:20, 1:40 and 1:80) significant reduction in the generation of Abs to both HLA-B*07:02 and B*81:01 [111], as compared with that of control media and mouse IgG control. The binding affinity of mAb TFL-007 is towards β2m-free HCs (Face-2) and not towards intact HLA (Face-1). This cannot be construed as a general effect since it involves specific binding of the mAbs to the cryptic domain exposed on Face-2. Since IVIg possess reactivity to several HLA antigens with a common shared sequence, IVIg may also bind to the shared epitopes to induce suppression of antibody production by the selected hybridoma.

#### 5.2.3. The Polyreactive mAbs (TFL-006/TFL-007) Mimic IVIg in Upregulating CD4+/CD25+/Foxp3+ Regulatory T-Cells (T-Regs)

The CD4+CD25+(IL-2Rα)Foxp3+ regulatory T-cells (T-regs) in humans and mice, though different in their cell surface markers, are capable of suppressing proliferation and cytokine production by CD4+CD25− T-cells in response to a variety of stimuli in vivo [146]. Suppression of proliferation and activation of CD8+ T-cells in vitro by human T-regs is mediated by “a cell-contact-dependent cytokine-independent mechanism” [147,148]. IVIg is known to upregulate T-regs isolated from patients with Kawasaki disease and common variable immunodeficiency diseases but not those isolated from healthy individuals [149,150]. The ability of TFL-006 and TFL-007 to induce proliferation of CD4+CD25+Foxp3+ T-regs obtained from normal and healthy donors was assessed [113]. The impact of IVIg was compared with that of the mAb TFL-007 on untreated and PHA-treated isolated fractions of CD3+/CD4+ human T lymphocytes. A variety of cell-surface markers, including CD4, CD25 (IL-2Rα), CD45RA, and Foxp3, were monitored using their respective diagnostic mAbs. We have compared the efficacy of IVIg and anti-HLA-mAbs TFL-006 and TFL-007 to induce proliferation of CD4+CD25+ (IL-R2α), CD34RA and Foxp3 T-regs obtained from normal and healthy donors [134]. The effect of different commercial preparations (GamaSTAN, Octagam and Gamunex) of IVIg (at 1:10 and 1:80) and TFL-007 (at 1:10 and 1:60) were studied on the PHA-untreated T-regs obtained from a healthy non-alloimmunized volunteer. 

Interestingly, all the three IVIg preparations did not significantly upregulate T-regs but in striking contrast, TFL-007 (at 1:10 but not at 1:60) significantly (*p*^2^: <0.001) upregulated T-reg cells. This finding once again resolves the role played by the polyreactive mAb binding to Face-2. Evidently, Face-2 is not overexpressed on normal T-cells but may be present at low levels, as observed for HLA-F and occasionally for HLA-B and HLA-C.

## 6. The Major Players in HLA-I-Mediated Cellular and Humoral Immune Homeostasis

### 6.1. HLA-I Variants as Receptors in Immune Regulation

HLA-I mediated immune homeostasis should be considered as a delicate and regulated balance of activation and suppression of immune and associated cells. It is also considered as a strategy to restore normalcy from alterations imposed by pathological conditions. Until 1989, β2m-associated HLAs (Face-1) were considered as the only functionally active cell-surface HLA class-I molecules. The Face-1 HLA-I molecules were known to present antigens derived from intracellular infection caused by pathogens to CD8+ cytotoxic T-cells (CTLs), which express TCRs specific for HLA-I. The T-cell co-receptor CD8 binds to the α3 domain of HCs of Face-1, while α1 and α2 domains of the HCs hold antigen peptides in the groove and present them to the receptor on T-cells (Figure 1). The antigen-presentation to CTLs initiates the cytolytic killing of the infected cells or even tumor cells. Since all human nucleated cells are potential targets for infections and malignancy, the HLA-I isomers are expressed on all such cells. 

The discovery of naturally occurring HLA-I variants expands the functional role of HLA-I in inflammation, tissue injury, infection, malignancy, and end-stage organ diseases. Thus, the discovery of Face-2 in activated immune cells revolutionized concepts of HLA-mediated immune homeostasis. The Face-2 variant of HLA-I has exposed amino acid sequences known to be masked by β2m (Figure 5). Interestingly, most of these sequences (Table 3, Figure 2, Figure 3 and Figure 4D–F) are highly shared among all HLA-I alleles of different isomers, namely HLA-A, HLA-B, HLA-C, HLA-E, HLA-F and HLA-G. 

Recent reports on Face-2 prompted attention to other HLA variants (homodimers, Face-3, and heterodimers, Face-4) emanating from Face-2. The cysteine residues on the open heavy chain of Face-2 can interact with the cysteine residues (at positions 64, 101 and 164) on the other allele’s heavy chain resulting in the genesis of Face-3 (Figure 6). Face-2 conversion into Face-3 or Face-4 on the cell-surface has the functional significance of masking the shared epitopes exposed by the paucity of β2m. Since these shared epitopes play a major role as receptors for several ligands, including Abs that bind to them (such as the mAbs TFL-006 and TFL-007), the immunohomeostatic roles of these HLA-I structural variants (Face-2, Face-3, Face-4, including that of Face-1) become explicit. Thus, Face-3, the homodimeric variant of Face-2, and Face-4, the heterodimeric variant of Face-2, have become important factors that can prevent the exposure of cryptic domains. They perform the function previously performed by β2-m in Face-1 HLA-I (Figure 14). Masking of the exposed shared epitopes seems to be an important step to regulate the signal transduction leading to activation of immune cells. However, elucidating the roles of Face-2, Face-3, and Face-4 in different pathological conditions may unravel their delicate and regulated role in immune homeostasis. More focused investigations in this direction are needed to understand the immunohomeostatic role of different Faces of HLA-I. 

### 6.2. Antibodies Formed against Face-1 and Face-2 as Major Players in HLA-Mediated Immune Homeostasis

#### 6.2.1. Profiles of HLA Abs in Sera of Normal Females and Non-Alloimmunized Males

Both IgM and IgG occur naturally in the sera of normal healthy males and females (Table 4 and Figure 7A–C). These Abs are formed against almost all alleles of HLA-A, HLA-B, HLA-C, HLA-E, HLA-F and HLA-G. The profiles of HLA Abs differ among individuals. Some normal healthy individuals have Abs against almost all HLA alleles, while some have Abs against only a few alleles (Table 4). Figure 7A–C illustrate the relative proportions (measured as mean fluorescence intensity or MFI) of IgM and IgG reacting to HLA-E, HLA-F, and HLA-G in the unpurified sera of the normal individuals. Many Abs escape detection due to presence of either anti-idiotypic antibodies or other factors contributing to the formation of immune complexes, as evidenced from the eluates of sera after purifying the Abs with Protein-G affinity columns (Table 5). It is clarified from the works of Morales-Bunerostro et al. [32] that serum naturally occurring anti-HLA Abs react to both Face-1 and Face-2 of HLA-I. However, very little is known about the functional roles of these naturally occurring sera Abs. To elucidate the functions of such anti-HLA Abs, investigators have used most often the mAb directed against Face-1, namely W6/32. Since the natural anti-HLA-I Abs also react well with Face-2, there is a need to examine the functional potentials of Abs directed against Face-2. 

#### 6.2.2. Functional Potentials of Antibodies Directed against Face-2 as Evidenced from the Observations Made on IVIg

Anti-HLA-I Abs observed in IVIg, pooled and purified from the plasma of thousands of individuals, represent the Abs found in normal healthy individuals. The profiles of anti-HLA Abs in IVIg are not comparable to normal native sera but to the sera purified by Protein-G affinity column. Figure 9 illustrates the profile of HLA-Ia reactivity of a commercial preparation of IVIg. The Abs were monitored using three kinds of beadsets, namely regular beadsets coated with Face-1 admixed with Face-2, beadsets treated with a mild acid to expose cryptic epitopes resulting in only Face-2, and iBeads carrying only Face-1. It is evident from the figure that IVIg contains Abs reacting to both Face-1 and Face-2. When IVIg was used in vitro on activated T-cells, proliferation was not enhanced, as was observed with the mAb W6/32, but rather blastogenesis and proliferation were significantly suppressed. This finding prompted further investigation on the functional potential of mAbs directed against Face-2.

HLA variants may play a major role in conjunction with the Abs they generate. Face-2 variants of HLA-I can elicit different kinds of epitope-specific Abs, due to exposure of several cryptic epitopes. The different kinds of Abs may include different grades of HLA-I polyreactive Abs ranging from the lowest to the highest polyreactivity (Table 6), depending on the frequency of the respective epitopes among HLA-I alleles [104]. Typical example of highly polyreactive Abs include TFL-006 (Table 7) and TFL-007. In addition, Face-1 is also capable of generating Abs that can bind simultaneously to β2-m and α-HC of HLA-I. A typical and most popular example is mAb W6/32, although as noted above, Tran et al. [103] raised unique concerns about this mAb’s affinity to the HLA-B Face-2. Despite the concerns, the very ability of W6/32 to bind to both β2-m and α-HC of HLA-I clearly identifies as a Face-1 directed Ab, while TFL-006 and TFL-007 remain as Face-2 specific polyreactive Abs. 

This review is restricted to one major issue of immune homeostasis related to activation CD3+/CD4+/CD8−, CD3+/CD4−/CD8+ and CD3+/CD4+/CD25+/Fox P3+ lymphocytes. An overview of HLA-I mediated immune homeostasis is diagrammatically illustrated in Figure 13 and Figure 14. They are enumerated as follows:(1)The difference in HLA variants found in normal and activated CD3+ lymphocytes. In normal healthy individuals, CD3+ lymphocytes express intact HLA-I (Face-1). The lymphocytes may get activated naturally upon inflammation and injury under different pathological conditions. In addition, other exogenous agents can induce activation, such as mAb 9.1 [125,126,127,128], phorbol myristate acetate (PMA), anti-CD3 antibody, phytohemagglutinin (PHA), brefeldin A (PFA), and IFN-γ [15]. Such activated T-cells initiate activation of transcription factors [64], production of cell-surface molecules such as IL-2Rα (Figure 13) [45,46,47], and Face-2 of HLA-I [12,15,16,17,115,116,117,118,119,120,121].(2)This unique shift in the cell-surface expression of Face-1 HLA-I on normal cells to Face-2 upon activation exposes epitopes that remain cryptic in Face-1 due to masking of amino acid sequences in the α2 domain by β2m.(3)This domain (Figure 2, Figure 3, Figure 4D–F Figure 5) exposes several sequences shared by HLA-A, -B. -C, -E, -F and -G. (Table 3). When all HLA alleles of all HLA isomers are seen on one screen, it is easy to pinpoint which sequences were cryptic and how they are shared by almost all alleles of all HLA isomers. The uniquely shared epitopes are important to understand and appreciate the role of HLA variants in immune homeostasis.(4)The exposure of cryptic epitopes may involve immediate immune recognition resulting in the formation of Abs against those shared and cryptic epitopes. This is exemplified by TFL-006 and TFL-007 and by their role in several immunoregulatory functions. Essentially, they are capable of suppressing proliferation of T and B-cells while promoting proliferation of T-regulatory cells (Figure 14).(5)Figure 13 clarifies the events that occur after the binding by mAbs TFL-006 and TFL-007 to Face-2 on the cell-surface of activated T-cells. The mAb binding to the exposed cryptic epitopes on Face-2 may involve a reversal of activation of T-cells mediated by signal transduction. In support of such an inference, it is documented that the elongation of the cytoplasmic tail of the HLA class-I open conformers will expose otherwise cryptic tyrosine^320^ [137,138,139] and serine^335^ [140] with a provision for phosphorylation (Figure 13). Although serine_335_ is generally considered the primary site of phosphorylation in this tail, phosphorylation of tyrosine^320^ has been documented by others [141,143]. Either way, what is most important is that the open conformers in activated normal human T-cells are associated with tyrosine phosphorylation and can enable *cis* interactions with cell-surface receptors or other signalling molecules [141,142,143,144]. The phosphorylation mediated by TFL mAbs may result in dephosphorylation of the cytoplasmic tails of CD3 molecules by activating phosphatases, leading to the arrest of transcription factors and synthesis of the proteins involved in blastogenesis and mitosis.(6)Again, a homeostatic function of the HLA variant Face-2 is the transient exposure of unique shared epitopes which is terminated by homo- and/or heterodimerization. Face-3, the homodimeric variant of Face-2, and Face-4, the heterodimeric variant of Face-2, recapitulate the function previously performed by β2-m in Face-1 HLA-I (Figure 14). The exposure of cryptic domains is blocked from receptor-ligand interactions and recognition by immunoregulatory Abs such as TFL-006 and TFL-007.

#### 6.2.3. Are Immunoregulatory Roles Restricted Only to HLA-polyreactive mAbs?

HLA-I isoform reactivity of the monoclonal Abs generated from immunization with two alleles (HLA-E^R107^and HLA-E^G107^) of recombinant Face-2 resulted in several mAbs with a wide range of allelic reactivity against HLA-Ia and HLA-Ib isomers (Table 6). In contrast to polyreactive mAbs such as TFL-006 and TFL-007, highly monospecific mAbs against HLA-E were also observed. The recognition of HLA-E-specific amino acid sequences by mAbs is the defining criteria of the monospecificity. There is dosimetric inhibition of HLA-E binding of the monospecific mAb TFL-033 by HLA-E-restricted amino acid sequences (^65^RSARDT^70^ and ^154^AESADNSKQES^144^). Such monospecific anti-HLA-E mAb is highly valuable for immunodiagnosis of HLA-E [105,106] on human cancers which are known to upregulate HLA-E gene expression correlated with overexpression of cell-surface HLA-E [151,152,153,154,155,156,157,158,159,160]. HLA-E gene expression in some cancers (e.g., melanoma) is ranked 19th among the most overexpressed genes [153,154,155,156,157,158,159,160,161,162]. HLA-Ib overexpression, concomitant with loss of HLA-Ia molecules in cancer cells, is correlated with disease progression and poor survival of patients [163,164,165,166,167]. The disease progression is attributed to the suppression of the tumor-killing activity of CD8+ cytotoxic αβ T-cells (CTLs) and natural killer (NK)T-cells [168]. Binding of HLA-E to the inhibitory receptors CD94 and NKG2A on both CD8+ CTLs and NKT-cells hindered the cell-mediated anti-tumor activity. The interaction between HLA-E and the inhibitory receptors involves the binding of amino acids located on the α1 and α2 domains of HLA-E to specific amino acids on CD94 and NKG2A [168]. This interaction is attributed to the loss of the anti-tumor activity of CD8+ CTLs as well as that of NKT-cells [169]. 

The specific sequences located on the α1 and α2 domains of HLA-E are indeed the sequences recognized by the monospecific mAb TFL-033. The amino acids on the α1 and α2 domains of HLA-E—respectively, (R^65^D^69^) and (S^152^)—are essential for recognition by the inhibitory receptors CD94 and NKG2A present on the cell-surface of CD8+ T-cells and NK cells [105,106,107,108,109,134,168,169]. Therefore, the mAb TFL-033 has the potential to block the ligands of inhibitory receptors on the α1 and α2 helices of HLA-E [107,108,109]. When the mAb TFL-033 was tested in vitro on isolated CD3+/CD8+ T-cells both in the presence and absence of PHA, the mAb triggered proliferation of both non-activated and PHA-activated CD8+ T-cells [111]. Importantly, a significant increase in the number of CD8+ T lymphoblasts was observed even in the absence of an activating co-stimulant (PHA). This unique property of augmenting the proliferation of CD8+ T-cells, particularly when the classical HLA-Ia antigens are downregulated concomitant with the upregulation of HLA-E, suggests the unique immunoregulatory potential of the HLA-E monospecific mAb TFL-033. 

These findings necessitate the need to further investigate the immunoregulatory potentials of other monospecific mAbs of HLA-F and HLA-G, after critical assessment of monospecificity by dosimetric peptide inhibtion studies. Such future investigations can expand the role of HLA further in immune homeostasis.

## 7. Conclusions

HLA class-I is long known to be a heterodimer with a polypeptide HC complexed with β2m on the surface of all nucleated cells. In the past three decades, it was discovered that there are three HLA-I variants (Face-2, Face-3 and Face-4) without β2m. Although the presence of these variants has been reported on activated immune cells, β2m-free HLA-I was observed in the monocytes of spondyloarthritis patients [170] and notably, the first trimester human endovascular trophoblast-cells [171] as well as on the extravillous cells of human placenta [172], which express both HLA-C and HLA-G. Siegel et al. [173] summarized reports on binding of Face-2 of HLA-C and HLA-F to activating and inhibiting receptors, such as KIR on NK cells. Increased frequency of HLA-C variants in rheumatoid arthritis (RA) is correlated with the strong association between RA and the KIR2DS2 activating receptors. They have also indicated that the homodimerized Face-2 (Face-3) can interact with the KIR receptors. HLA-C variants are also associated with increased susceptibility to systemic lupus erythematosus [174]. Activating KIR2DS4 was shown to specifically recognize Face-2 of HLA-F [20]. Infrequent reports of their presence of these variants in some, if not all, non-immune and malignant-cells have appeared [24,25,26,170]. It was unknown whether there are Abs directed against the three β2m-free HLA-I variants (Faces 2–3), until Morales-Bunerostro et al. [32] reported that the HLA-I Abs found in normal non-allo immunized males and females are against Face-2 variants of HLA-I. Indeed, many de novo donor-specific antibodies recognize β2m-free HLA but do not recognize intact HLA heterodimers [171].

Commonly used anti-HLA-I mAb W6/32, binds to both HLA-I HC and β2m. The assumption that the mAb W6/32 is specific for Face-1 was shattered when Tran et al. [103] reported that the mAb W6/32 recognizes an epitope preserved in free nonreduced αHCs of most HLA-B antigens but not in other HLA class-I α HCs. Essentially identical results were also obtained with another pan-HLA class-I mAb MEM-147 which cross-blocks W6/32. It is yet to be examined whether W6/32 recognizes an epitope on αHCs of HLA-Ib molecules.

The Face-2 HC-exposed epitopes that were previously masked by β2m are, interestingly, shared by all HLA-I isomers. Abs binding to these shared epitopes are considered as polyreactive mAbs. Immunizing the Face-2 of recombinant HLA-E elicited several Abs, some of them are truly polyreactive (TFL-006 and TFL-007). In contrast to mAb W6/32, they suppressed proliferation and blastogenesis of activated CD4+ T-cells and CD8+ T-cells but enhanced proliferation of T-regulatory cells (Figure 14). Face-2 also elicited Abs directed against specific domain of HLA-E not shared by any other HLA-Isomers. One such Ab is TFL-033, which has induced proliferation of both non-activated and activated CD8+ T-cells. 

In summary, there exist several steps in HLA-I mediated immunohomeostasis.
(1)Exposures of cryptic epitopes in β2-m free HC (Face-2) of HLA upon activation of cells. Such activation may occur during inflammation, injury, malignancy, and other pathological conditions, such as arthritis, autoimmune diseases and cancer.(2)Several of these exposed cryptic epitopes are shared by almost all HLA-I isomers.(3)Such exposure invites immune recognition by Abs and various other ligands.(4)It appears that such exposure may not last long, and it is prevented by homo- or heterodimerization of Face-2.(5)The Abs generated by the exposed cryptic epitopes are uniquely polyreactive with Face-2 of all HLA isomers since the epitopes are shared by all HLA isomers.(6)Binding of these Abs is capable of suppressing the proliferation and blastogenesis of CD4+ and CD8+ T-cells.(7)Binding of these Abs enhances the proliferation of T-regulatory cells, which are capable of several immunoregulatory functions.(8)The function of these Abs directed against cryptic epitopes on Face-2 is different from that of mAb W6/32 which is capable of binding to Face-1 of all loci (and Face-2 of B alleles) and promoting the proliferation of immune and non-immune cells.

This review highlights the importance of both polyreactive as well as the monospecific HLA-I mAbs as key elements for maintaining immune homeostasis. Confirming the epitope’s specificity of the mAbs by dosimetric peptide inhibition is a necessary prerequisite for understanding their precise roles. Examining the HLA alleles and their shared epitopes, as well as their corresponding Abs formed during specific pathological conditions, would enable developing therapeutic and diagnostic mAbs. These mAbs will have potential usefulness in specific immunodiagnosis and for developing active or passive immunotherapy for a variety of pathological conditions such as cancer, endstage organ diseases, and autoimmune diseases.

## Figures and Tables

**Figure 1 antibodies-11-00058-f001:**
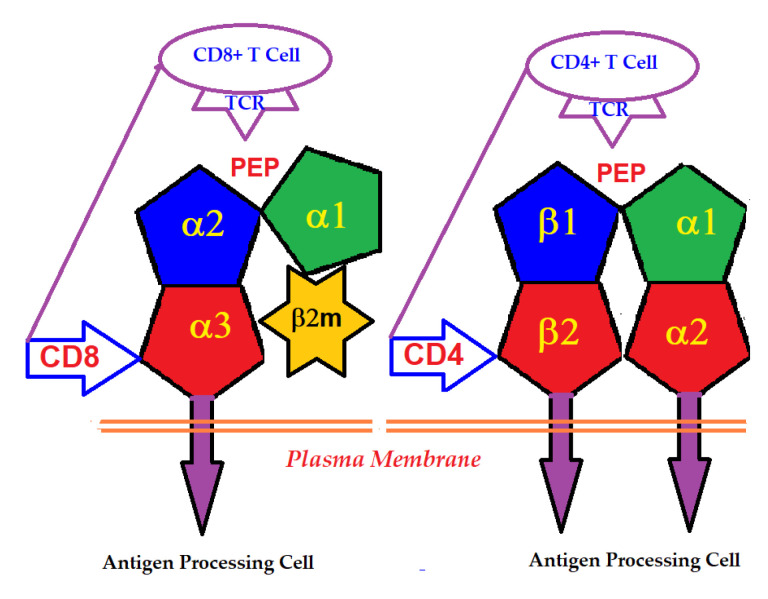
Binding of T-cell co-receptors CD8 and CD4 to α3-domain of HLA-I and β2-domain of HLA-II, respectively.

**Figure 2 antibodies-11-00058-f002:**
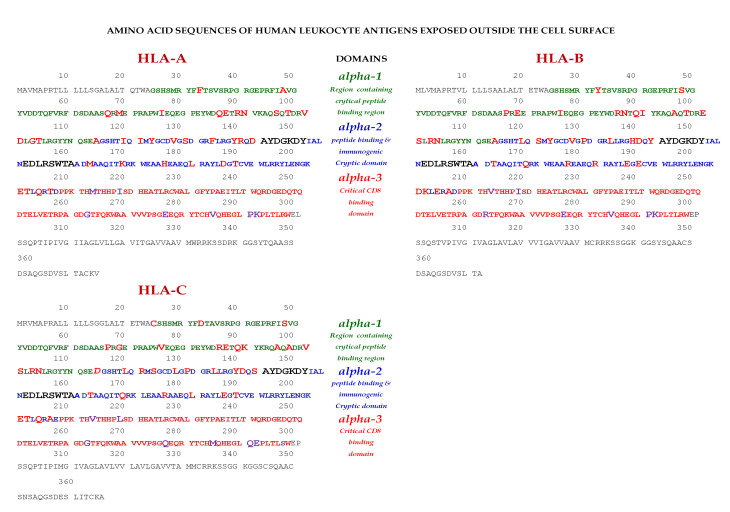
Exposed and cryptic (in α2 domain) amino acid sequences of classical HLA-I (HLA-Ia) on the cell-surface.

**Figure 3 antibodies-11-00058-f003:**
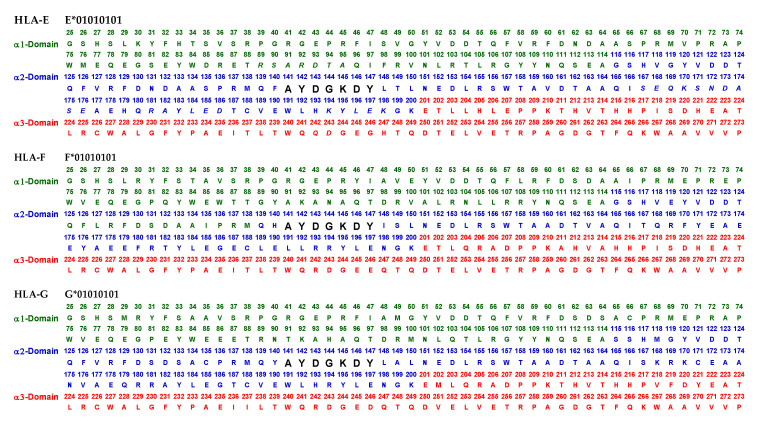
Exposed and cryptic amino acid sequences (in α2 domain) of non-classical HLA-I (HLA-Ib) on the cell-surface.

**Figure 4 antibodies-11-00058-f004:**
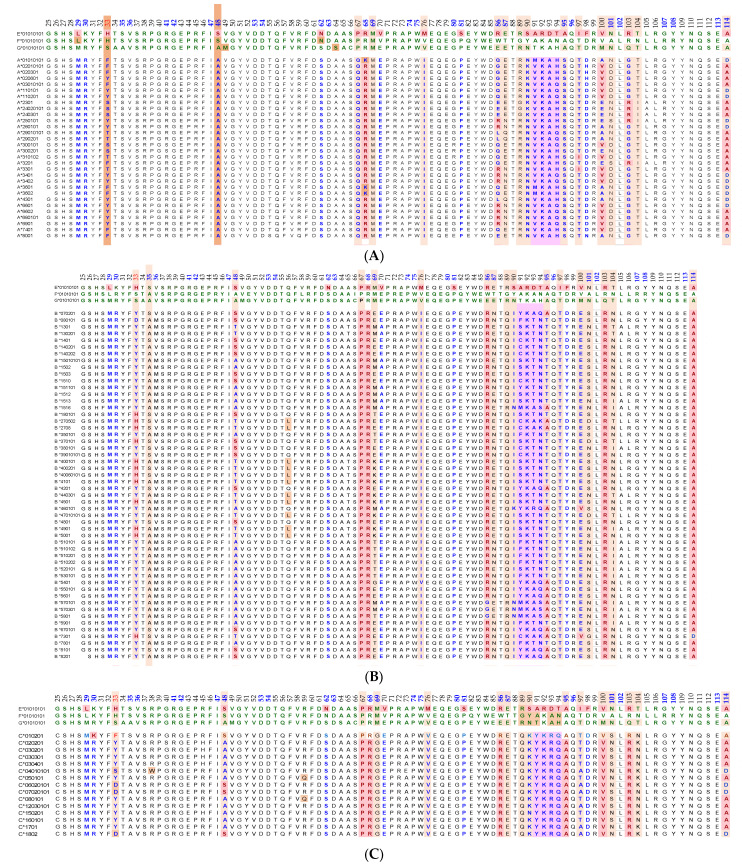
An overview of the polymorphic amino acid sequences of α1 and α2 domains in the classical (HLA-Ia: HLA-A/-B/-C) and non-classical (HLA-Ib: HLA-E/-F/-G). Essentially, α1 domain of HLA isomers carry monospecific sequences characteristic of each allele. This figure illustrates a specific sequence of HLA-E, namely ^90^SARDTA^95^ in the α1-domain, which is restricted exclusively to HLA-E. This sequence is recognized by the mAb TFL-033 and the isolated sequence inhibited the mAb binding to HLA-E. Similarly, a most commonly shared amino acid sequence (^141^AYDGKDY^147^) located in α2-domain of all alleles of HLA-Ia and HLA-Ib is illustrated. Although the number of lleles of HLA-I isomers exceed several thousands as presented in Table 1, this figure is only limited to alleles of HLA-Ia and Ib commonly used to monitor antibodies using antigen coated beadsets on Luminex platform. Interestingly, further examination of several thousands of alleles revealed that the amino acid sequences of shared epitope (^141^AYDGKDY^147^) are present in all examined alleles. Sequences of all alleles or the presence of ^141^AYDGKDY^147^ in all alleles of all isoforms can be found in https://www.uniprot.org/uniprotkb/P04439/entry#sequences, accessed on 1 July 2022, or at https://ebi.a.c/uk/ipd/im/hla/, accessed on 1 July 2022. (**A**) HLA-A α1-domain; (**B**) HLA-B α1-domain; (**C**) HLA-C α1-domain; (**D**) HLA-A α2-Domain; (**E**) HLA-B α2-Domain; (**F**) HLA-C α2-domain.

**Figure 5 antibodies-11-00058-f005:**
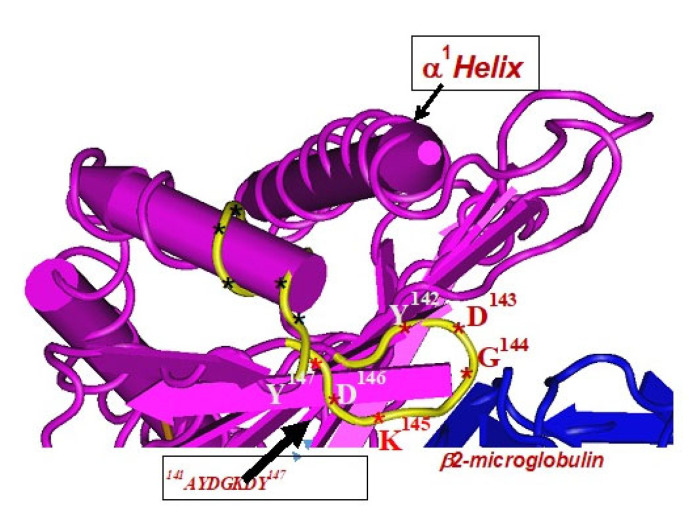
The amino acid sequences of α2-domain of HLA-I shared by HLA-A, B, C, E, F and G loci, that are hidden (cryptic domain) by β2-microglobulin. These sequences are highly immunogenic. For details see Table 3.

**Figure 7 antibodies-11-00058-f007:**
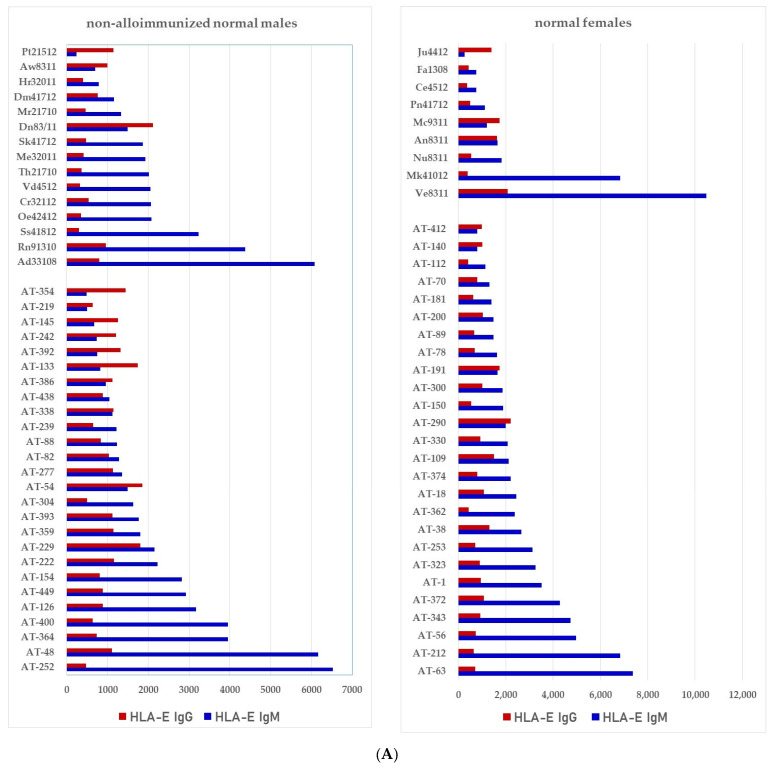
Documentation of IgM and IgG antibodies against HLA class Ib molecules in the sera of male (non-alloimmunized) and female (volunteers). Interestingly, naturally occurring IgM antibodies are more prevalent than IgG for HLA-E (**A**). In contrast, naturally occurring IgG antibodies are more prevalent than IgM for HLA-F and HLA-G (**B**,**C**). Simultaneous prevalence of higher IgM than IgG in HLA-E, HLA-F and HLA-G, suggests that these individuals may be experiencing induction of IgM due to various pathological conditions such as inflammation, injury and other disorders. (**A**) Anti-HLA-E IgM and IgG; (**B**) PosiAnti-HLA-F IgM and IgG; (**C**) Anti-HLA-G IgM and IgG.

**Figure 8 antibodies-11-00058-f008:**
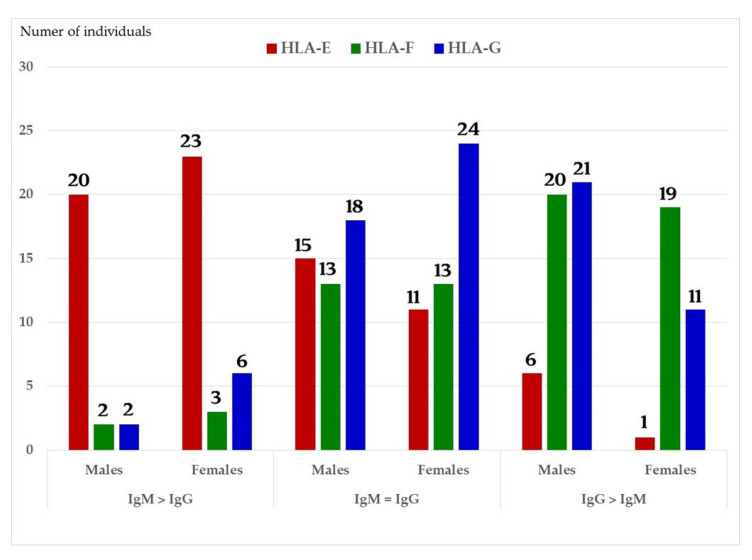
The relative ratios of IgM and IgG in the sera of normal males (non-alloimmunized) and females. Note the high prevalence of IgM Abs against HLA-E in a number of males and females, and high prevalence of IgG Abs against HLA-G.

**Figure 9 antibodies-11-00058-f009:**
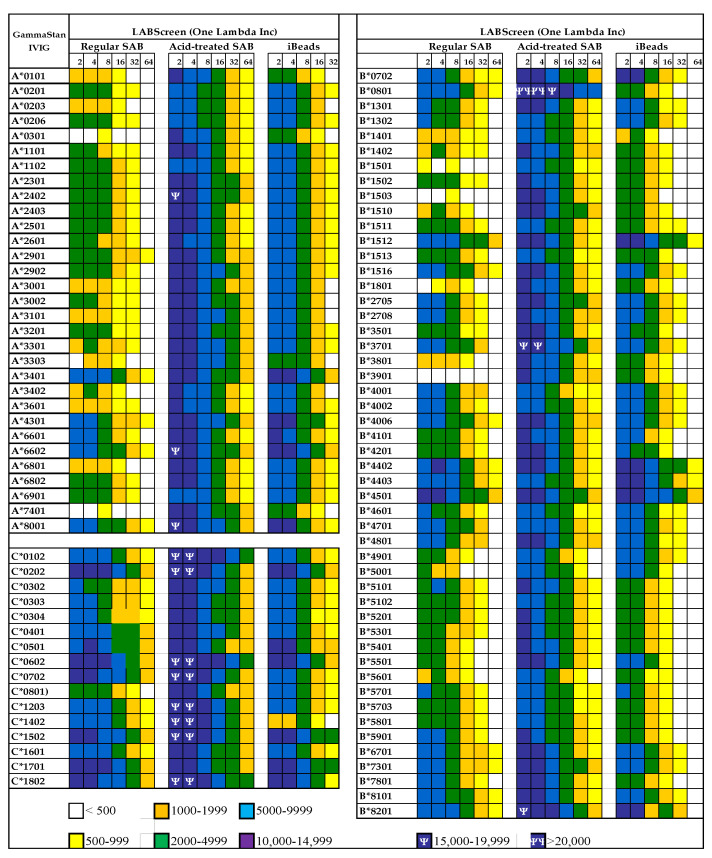
Documentation of antibodies against HLA class Ia molecules (HLA-A, HLA-B and HLA-C) in a commercial preparation (GamaStan) of Intravenous Immunoglobulin (IVIg). * Suggesting that the individuals may possibly carry naturally occurring antibodies formed against most commonly shared epitopes. The profiles of IgG antibodies were obtained on three different kinds of beadsets (LABScreen, One Lambda Inc, Canoga Park, CA). The different kinds of beads are (1) Regular LABScreen single antigen beadsets (SABs) which are coated with Face-1 admixed with Face-2 or open conformers, (2) the same SABs after mild acid treatment resulting in the exposure of cryptic epitopes, resulting in molecules similar to naturally occurring Face-2 on activated immune cells, (3) iBeads, the same regular LABScreen SABs which are gently trypsinized to remove Face-2, to leave only Face-1 on the solid matrix. The difference in the profile between regular beads and iBeads also illustrates the admixture of Face-1 and Face-2 in regular LABScreen SAB commonly used in the clinical laboratories. If the objective of the clinicians is to monitor antibodies against Face-1, then only iBeads or Immucor Beadsets will serve the specific purpose. Mean Fluorescent Intensities (MFI) are indicated in different colors and symbols.

**Figure 10 antibodies-11-00058-f010:**
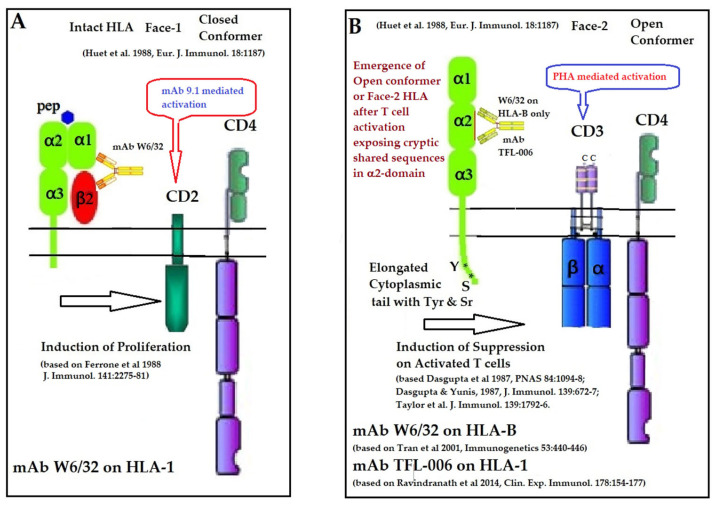
Illustration of the role of mAb W6/32 in enhancing the CD2 and CD3 mediated activation of T-cells (**A**) as well as the role of HLA-I polyreactive mAbs TFL-006 and TFL-007 in suppression of (PHA)-CD3-mediated activation of T lymphocytes (**B**) [125,126,127,128,129,130,131].

**Figure 11 antibodies-11-00058-f011:**
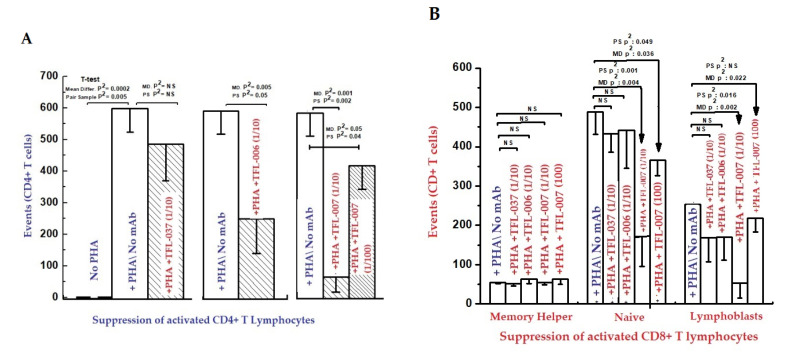
Inhibition of Phytohaemagglutinin (PHA)-activated T-lymphocytes in vitro with two ((**A**) CD4+; (**B**) CD8+).

**Figure 12 antibodies-11-00058-f012:**
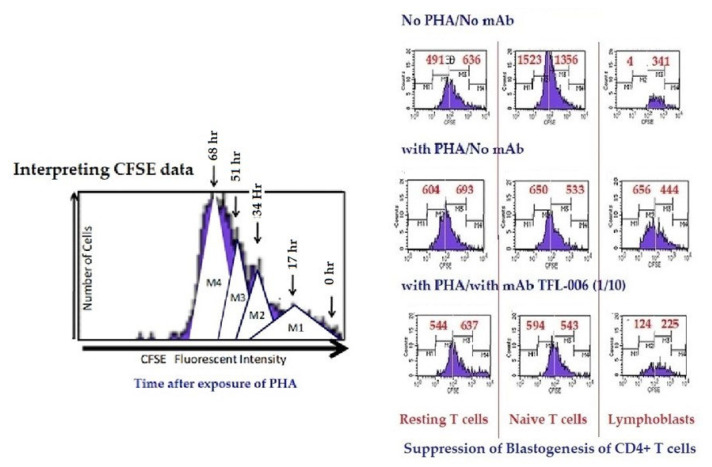
Suppression of in-vitro proliferation of PHA-activated CD4+ T lymphocytes by mAb TFL-006s (culture supernatant). The carboxyfluorescein succinimidyl ester (CFSE)-labelled lymphocytes were cultured with or without PHA or with PHA and mAb TFL-006s at 1/10 dilution. Three days after culture, cells were labelled with fluorescent dye-conjugated anti-CD4+ or anti-CD8+ Abs before analysis. CFSE labelling allowed us to gauge and show cell proliferation: when the CFSE-labelled cell population undergoes mitosis, after 72 h it has migrated from the right to the left side of each rectangular box (as shown in the top figure) depending on the number of mitoses. The distance moved shows the number of cell divisions. The effects of mAb TFL-006s on the proliferation of CD4+/CFSE+ T lymphocytes are shown in each box. After incubating cells with CFSE, the cells were treated as noted. Each box in the figure is divided by a vertical line into two sub-boxes, the right for mitoses 1 and 2 (M1/2) (parent lymphocytes) and the left for mitosis 3 to 5 (M3–5) (the progeny). The cell number after each treatment (including ‘no PHA’) of each T lymphocyte population was counted and compared, the number shown in each sub-box. Note that with no PHA the number of cells in the M3–5 sub-box is very meagre for all groups of CD4+ T lymphocytes. With PHA-only treatment, the very high number of cells for M3–5 indicates that proliferation has occurred in all three groups. The impact of treatment with PHA and TFL-006s is striking: the number of cells in the M3–5 sub-box is reduced, indicating suppression of proliferation. No such decrease was observed with resting or naive T lymphocytes.

**Figure 13 antibodies-11-00058-f013:**
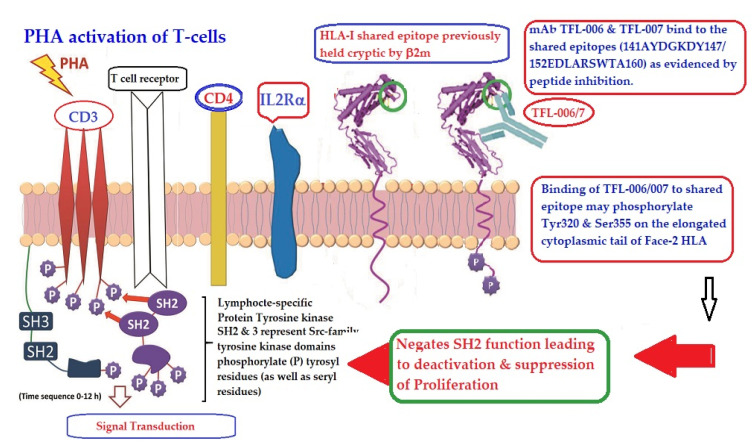
An illustration of the possible mechanism of suppression of PHA activated T-cells mediated by mAbs directed against shared epitopes exposed in the Face-2 of almost all alleles of HLA-I. The figure shows CD3/T cell receptor (TCR)/CD4 on the bi-layered lipid membrane on the T-cells upon activation (Yellow complex arrow). Phosphorylation of the cytoplasmic domain of CD3 is induced by Lymphocyte-specific protein tyrosine kinase (LCK). SH represents Scr-family LCK. They phosphorylate the tyrosyl (even seryl) domain of the cytoplasmic tail of CD3. This event leads to activation of transcription factors and transcription cell-surface molecules such as IL2Rα and Face-2. Importantly, the HLA-I open conformers devoid of β2m (Face-2) expose unique epitopes shared by almost all HLA-I alleles. It is the site recognized by TFL-006 and TFL-007. It is postulated that the binding of these specific mAbs to the epitopes on the Face-2 may initiate phosphorylation of the elongated cytoplasmic tail. The elongated cytoplasmic tail results in the exposure of cryptic tyrosyl residue at position 320 [137,138,139] and serine at position 35 [140]. This leads to transduction that initiates dephosphorylation of the cytoplasmic domain of CD3, resulting in suppression of activation of T-cells. Evidently, the T-cell activating CD3-phosphorylation can be reversed.

**Figure 14 antibodies-11-00058-f014:**
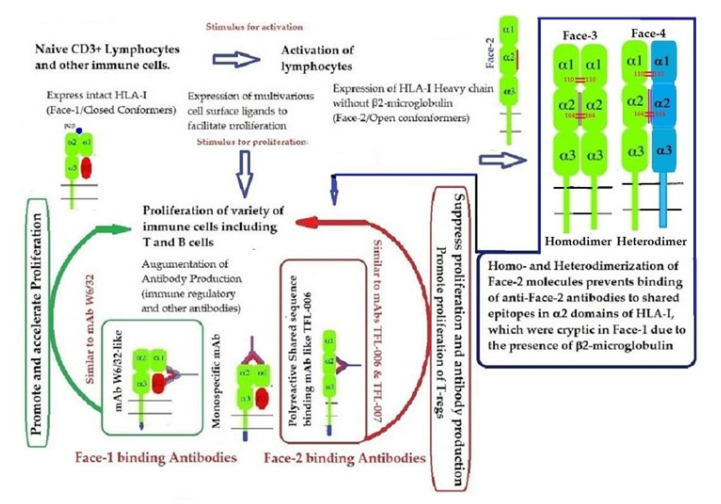
Graphic overview of the roles played by HLA-I structural variants (Face-1, Face-2, Face-3, and Face-4) together with the Abs directed specifically against Face-1 (represented by the mAb W6/32) and/or the Abs directed specifically against Face-2 (exemplified by the mAbs TFL-006/007) in HLA-I mediated immune homeostasis.

**Table 1 antibodies-11-00058-t001:** The diversity of α-chain of HLA-I and α- and β-chains of HLA-II is exemplified by the number of the named alleles and proteins (Based on December 2021, IPD-IMGT/HLA Database) ^1,2^.

HLA Class I Alleles			HLA Class II Alleles					
α-Chain			α-Chain			β-Chain		
Genes	Alleles	Proteins	Genes	Alleles	Proteins	Genes	Alleles	Proteins
A	7452	4355	DRA	32	5	DRB1	3196	2152
B	8849	5343				DRB2	1	0
C	7393	4095				DRB3	423	314
E	310	121				DRB4	215	139
F	50	7				DRB5	171	131
G	102	35	DQA1	442	205	DQB1	2230	1407
			DQA2	40	11			
			DPA1	406	173	DPB1	1958	1223

^1^ Based on 2022 report found in ebi.ac.uk/ipd/imgt/hla/about/statistics/ (accessed on 1 July 2022); ^2^ Based on HLA Nomenclature @hla.alleles.org (accessed on 1 July 2022).

**Table 2 antibodies-11-00058-t002:** HLA Class I and II isoforms and the shared sequences in the α3-domain in HLA-I and β2 domain HLA-II that bind to T cell co-receptors.

HLA-I Isoforms and Domains																						
				**α1-Domain**							**α2-Domain**						**α3-Domain**					
A				25–114							115–200						201–298					
B				25–114							115–200						201–298					
C				25–114							115–200						201–298					
**Amino Acid Sequences Shared by Almost All Known Alleles of Hla-A, -B, -C, -E, -F and -G, and Their Positions in CD8 Binding α3-Domain**																						
^222^	E	A	T	L	R	C	W	A	L	G	F	Y	P	A	E	I	T	L	T	W	Q	^242^
^244^	D	G	E	G	H	T	Q	D	^251^													
^253^	E	L	V	E	T	R	P	A	G	D	G	T	F	Q	K	W	A	^269^				
**HLA-II Isoforms and Domains**																						
				**α1-Domain**				**α2-Domain**								**β1-Domain**				**β2-Domain**		
DRA				26–109				110–203				DRB				30–124				125–227		
DQA				24–119				120–203				DQB				30–121				122–215		
DPA				29–115				116–209				DPB				33–126				127–229		
**Amino Acid Sequences Shared by Almost All Alleles of HLA-DR,-DQ and DP & Their Positions in CD4 Binding β2-Domain**																						
DRB							^150^	N	G	D	W	T	F	Q	T	L	V	M	L	E	^162^	
							^185^	T	V	E	W	R	A	R	S	E	S	A	Q	S	K	^198^
DQB							^150^	N	G	D	W	T	F	Q	T	L	V	M	L	E	^162^	
							^185^	T	V	E	W	R	A	R	S	E	S	A	Q	S	K	^198^
DPB							^148^	N	G	D	W	T	F	Q	T	L	V	M	L	E	^160^	
							^183^	T	V	E	W	R	A	R	S	E	S	A	Q	S	K	^196^

**Table 3 antibodies-11-00058-t003:** Most commonly shared and cryptic peptide sequences in the α2-domain of the heavy chains of all HLA-I isoforms (HLA-A/-B/-C/-E/-F and -G).

HLA-E Peptide Sequences	HLA Alleles					Frequency of Occurrence in HLA by Rank	Prediction Scores				Immunogenicity Rank
	Classical HLA-Ia			Non-Classical HLA-Ib			Method 1	Method 2	Method 3	Method 4	
[Total Number of Amino Acids]	A	B	Cw	F	G		Beta-Turn	Antigenicity	Flexibility	Hydrophilicity	
							Chou & Fasman [7]	Kolaskar & Tangaonkar [8]	Karplus & Schulz [9]	Parker, et al. [10]	
^141^AYDGKDY^147^ [11]	**491**	**831**	**271**	**21**	**30**	**1**	**1.204**	**0.989**	**1.061**	**4.243**	**1**
^150^LNEDLRSWTA^159^ [12]	239	219	261	21	30	2	1.046	0.983	1.039	2.443/2.329	2
^161^DTAAQI^166^ [6]	0	824	248	0	30	3	0.813	1.065	0.978	1.957	3
^187^TCVEWL^192^ [6]	282	206	200	0	30	4	0.841	1.115	0.929	−0.914	4

Amino acid number starting with 1–24 in the leader sequence. The data in bold are significant for this study.

**Table 4 antibodies-11-00058-t004:** Documentation of antibodies against HLA class Ia molecule in the sera of non-alloimmunized male volunteers. The profiles of antibodies against HLA-A, HLA-B and HLA-C differ among different individuals. Antibodies against different alleles of each HLA isomers differ as follows: HLA-C > HLA-A > HLA-B. One male (Ho) has antibodies against almost all alleles of HLA-Ia tested suggesting that the individual may possibly carry naturally occurring antibodies formed against most commonly shared epitopes such as that of ^141^AYDGKDY^147^ illustrated in Figure 4. No or low MFI in several individuals does not imply the absence of antibodies, as antibodies against almost all alleles can be seen after purification of IgG using the Protein-G affinity purification column as illustrated in Table 5. It implies that the naturally occurring HLA antibodies may be masked by other factors, such as anti-idiotype antibodies and immune complexes.

		Pt^GF^	Kt^F^	Pj^GS^	Mr^F^	Nr^S1^	Hr^S2^	Ho	Rn	Th	Sk	Vd	Me	Cr	Ad	Ra
	NCAlbumin				25			190	120	36	943	503	55	537	91	
	PC		788		791	1977		1317	1915			1274	1418		909	
Anti-HLA-A IgG	A*0101			811	856	1962	1372	3592	1326			880	1393	568	8337	2847
	A*0201					588	595	782	770						2798	1006
	A*0203					725		976	765						3576	1293
	A*0206							620	617						1959	610
	A*0301			2564				3091	882	502			3837		2258	558
	A*1101							1386							1413	
	A*1102			722		1003		3528	1005		2776	903			6486	2080
	A*2301					556		5166		1956			610	2914	3253	1071
	A*2402					588		5003		2331				3357	3602	978
	A*2403			513		809		3331	511	1984			1159	1465	5223	1429
	A*2501				1112	523		5187					3482	546	2767	2174
	A*2601					775		3042					1074	954	4224	2621
	A*2901							657	553						1361	534
	A*2902							506	593						968	
	A*3001					718		1103	605	1680					3314	1416
	A*3002							1021	525	590					2005	885
	A*3101					877		1743	586						3954	1632
	A*3201			2600				8422					1142		2692	577
	A*3301			864									2464	5584	914	511
	A*3303												3138		767	
	A*3401							1385					2810		1455	853
	A*3402							667					2179	1594	1386	535
	A*3601				508	785		5724	574						4266	1373
	A*4301							1489						506	1632	567
	A*6601							2102					1618		2506	2088
	A*6602							1021	810				671		2815	526
	A*6801							1426					3008	1374	1941	712
	A*6802							1372					587	585	2713	877
	A*6901			548	638	1579		2336	512				895	2050	6736	2171
	A*7401			917				2817							2176	
	A*8001		994	798	1733	1536		4700	1828	2457		683	609	1214	3009	1132
Anti-HLA-B IgG	B*0702								791		645		846			
	B*0801			605		1738		1449	2221		1068	503		6274	2380	1261
	B*1301							1143	663							
	B*1302							849								
	B*1401							670								
	B*1402							1235	506			555				
	B*1501							1144					581			
	B*1502							2104						4381		
	B*1503							1177								
	B*1510			597				1729				866	784	2226		
	B*1511			1365		1075		3617	1153	627	604	1441	990	1507	545	667
	B*1512					550		2487	768		1634	508	572	805	1149	
	B*1513			507				2356				643		1906		
	B*1516							3544								
	B*1801							1342	807					568		
	B*3501							1244						6375		
	B*3701							1137								
	B*3801							1720								
	B*3901							616								
	B*4001							582								
	B*4002							1095								
	B*4006							671	818							
	B*4101							774								
	B*4201								531		915					
	B*4402							994		590						
	B*4501							1412								
	B*4601			674		817		2203	1314	823	1104	503		1086	1982	740
	B*4701							669								
	B*4801							842				545				
	B*4901			747				4828					986			
	B*5001			710				1753					1091			
	B*5101			529				5536				678		1550		
	B*5102			556				5125				733	610	8066		
	B*5201							1844				694				
	B*5301			1013		676		2216	600			1218	702	2217		
	B*5401			688		553		1946	555			814	506		808	
	B*5501							853			514					
	B*5601			578				1020					730			
	B*5701							2892		11,023				1346	4190	
	B*5703							1821		7141				1143	3874	
	B*5801							1864		10,634				500	2602	
	B*5901			1022		1152		5069	916		651	1525	845	6396		705
	B*6701	796				543		517	1089		990					
	B*7301							1185					679	622		
	B*7801							3702						4296		
	B*8101							827	789	526			553			
	B*8201			707		570		1627	1439	715	1358	798	591	1140		
Anti-HLA-C IgG	C*0102	1219						733	920	1503		1287			1079	717
	C*0202			568		712		1160	1012	569	874	772				1088
	C*0302					717		916	1205	1516				556	972	833
	C*0303					1034		907	1248	1575				657	568	1414
	C*0304					712		932	1080	1567				666	503	799
	C*0401	1470		530		607		983	799	1049		689		844		1139
	C*0501							1146	778	551		803				
	C*0602			913		1001		1890	7846	1287	1458	1834	1137	866	1293	1553
	C*0702			841		1344		1666	1531	1162	1226	1307		580	808	1325
	C*0801							762	733			526	681			
	C*1203							1107	2244	601		1409	1188	696	639	
	C*1402			545		794		900	784	1345	658	949			561	1347
	C*1502	578						1027	3112	712	848	1314	1210	1391	743	
	C*1601					618		856	1856	539	535	801			608	900
	C*1701			933		1202		1696	1568	1651	677	1304		589	2376	1684
	C*1802			601		531		1210	2987	1578	678	1204		601		

GF: Grand Father; F: Father; GS: Grandson of T-family; F: father, S1: first son; S2: second son of R-family. * Suggesting that the individuals may possibly carry naturally occurring antibodies formed against most commonly shared epitopes.

**Table 5 antibodies-11-00058-t005:** HLA-Ia Ab profiles in the sera of normal non-alloimmunized male volunteers after purification of IgG using Protein-G affinity purification columns, elaborate the profile of the naturally occurring anti-HLA-I antibodies, suggesting that under natural conditions these antibodies may be masked by other factors, such as anti-idiotypic antibodies and immune complexes.

		Mr	Ho	Rn	Th	Sk	Vd	Me	Cr	Ad
	NC (Albumin)	862	2101	940	1652	5052	1451	874	2752	1326
Anti-HLA-A IgG profile after affinity purification	A*0101	631	1697	747	1255	1352	505	646	1895	804
	A*0201	661	1563	797	1209	1668		648	2172	878
	A*0203	712	1641	756	1155	1727		604	2064	886
	A*0206	789	1804	911	1324	2127	518	730	2650	965
	A*0301	833	2197	911	1587	1898	555	733	2408	1018
	A*1101	970	2645	1066	1639	2053	730	808	2892	1132
	A*1102	741	1684	1056	1277	1662	579	637	2044	949
	A*2301	897	2980	1133	1857	2007	688	1074	2841	1193
	A*2402	1196	2967	1299	2284	2576	776	1175	3558	1437
	A*2403	1016	2446	1108	1931	2261	687	894	2519	1238
	A*2501	702	1847	804	1207	1576	505	693	2322	942
	A*2601	1104	3401	1136	1976	2125	797	988	2721	1272
	A*2901	902	2911	1013	1720	1640	778	892	2870	1090
	A*2902	927	4136	1044	1781	2150	784	869	2832	1145
	A*3001	1063	2421	1080	1751	3524	702	891	2540	1227
	A*3002	686	1780	797	1286	1631	519	636	2077	896
	A*3101	656	2074	741	1164	1746		615	2351	801
	A*3201	779	3294	866	1523	1533	660	772	2249	966
	A*3301	994	2807	958	1761	2475	669	835	2668	1111
	A*3303	752	2229	806	1357	1804		646	2177	891
	A*3401	1141	3851	1162	2004	2741	858	1121	2746	1272
	A*3402	688	1675	778	1258	1546	513	663	2195	861
	A*3601	881	1977	924	1510	1848	652	750	2257	1010
	A*4301	1114	3782	1180	2081	2452	709	963	2308	1342
	A*6601	924	3247	935	1575	2620	615	785	2061	1066
	A*6602	983	3189	972	1759	3099	643	826	2226	1069
	A*6801	651	1427	705	1236	1587		673	1914	775
	A*6802	932	2216	1065	1555	2094	989	750	3338	983
	A*6901	1262	2777	1314	2237	2614	835	1138	2835	1521
	A*7401	875	2719	897	1519	1734	602	755	1802	1025
	A*8001	931	2885	1033	1676	2282	730	876	2141	1127
Anti-HLA-B IgG profile after affinity purification	B*0702	699	1549	825	1193	1792		657	2036	839
	B*0801	796	2020	890	1447	1650	556	703	2576	975
	B*1301	1331	3178	1356	2231	2725	817	1088	3103	1461
	B*1302	967	2500	1096	1639	2385	660	837	2697	1126
	B*1401	1198	2702	1231	2022	2203	950	1162	2876	1368
	B*1402	977	2439	1093	1797	2176	748	919	2324	1093
	B*1501	747	1602	860	1329	2026		718	2134	945
	B*1502	942	2610	1070	1764	2453	637	928	2745	1233
	B*1503	986	2277	1039	1724	2208	595	739	2588	1130
	B*1510	965	2934	1047	1714	2138	742	887	2357	1175
	B*1511	1443	3526	1473	2466	2989	1010	1169	3462	1567
	B*1512	751	2163	829	1496	2721		2523	3030	1147
	B*1513	1232	2917	1321	2105	2597	839	1097	2868	1455
	B*1516	1205	4241	1389	2398	2725	1360	1199	3158	1503
	B*1801	752	2024	853	1400	1641	565	658	2196	912
	B*2705	595	1378	681	1084	2021		589	1990	764
	B*2708	551	1514	651	932	1669		520	2107	711
	B*3501	940	2199	1003	1667	2216	608	864	3054	1195
	B*3701	1518	4054	1967	2613	2897	949	1259	3369	1540
	B*3801	1191	2110	1095	1706	2060	621	853	2778	1160
	B*3901	575	1291	657	1051	1446		520	1800	757
	B*4001	781	1980	910	1448	1761	520	698	3228	980
	B*4002	624	1669	713	1124	1313		575	2044	783
	B*4006	1354	3682	1403	2280	2773	1006	1109	3287	1511
	B*4101	760	1823	848	1413	1659	528	688	2299	940
	B*4201	694	1603	765	1160	2014		632	2121	857
	B*4402	1121	2455	1196	1885	2067	710	2587	2575	1230
	B*4403	811	2028	1044	1382	1972	515	2353	2474	932
	B*4501	756	1814	786	1325	1780	500	2709	2221	918
	B*4601	1090	2941	1466	2220	2396	898	1163	3297	1470
	B*4701	879	2687	994	1729	1867	670	823	2567	1064
	B*4801	1015	2291	1088	1769	1908	770	858	3066	1173
	B*4901	929	2200	978	1725	2046	580	820	2569	1168
	B*5001	675	1898	784	1207	1563	500	607	2152	880
	B*5101	1118	2580	1214	2079	2259	724	973	2494	1355
	B*5102	1116	2754	1140	1997	2271	842	976	2609	1341
	B*5201	1050	3355	1227	1934	2261	744	982	2391	1299
	B*5301	1251	3024	1359	2264	2474	845	1100	2725	1469
	B*5401	1011	2528	1101	1737	2264	748	859	2661	1192
	B*5501	955	2327	1020	1629	2089	599	806	2427	1111
	B*5601	907	2698	1025	1594	2244	553	843	2667	1186
	B*5701	762	2155	937	1512	2081	538	723	2144	990
	B*5703	995	2740	1171	1864	2345	746	953	2789	1218
	B*5801	1338	3218	1424	2331	3213	924	1185	2930	1523
	B*5901	1155	2730	1207	1960	2502	766	1002	2600	1377
	B*6701	661	1601	827	1180	1895	709	700	2412	841
	B*7301	842	2326	960	1590	2458	555	932	2132	1085
	B*7801	1150	2527	1186	1964	2544	743	948	2538	1346
	B*8101	910	2241	1117	1658	2130	649	790	3022	1100
	B*8201	2293	2886	1288	2311	2168	864	1006	2819	1445
Anti-HLA-C IgG profile after affinity purification	C*0102	1740	4894	1735	2972	2873	1221	1397	3892	1684
	C*0202	1941	4568	1894	3222	3936	1410	1504	3505	2011
	C*0302	1830	4391	1829	3077	3369	1169	1512	3621	1967
	C*0303	1663	3828	1602	2839	2851	1182	1409	3076	1849
	C*0304	1900	4334	1801	2974	3272	1234	1538	3437	1977
	C*0401	2166	6468	2126	3870	3863	1587	1897	3951	2305
	C*0501	1967	3806	1726	3015	3404	1216	1912	3560	1886
	C*0602	2165	4446	2175	3835	3672	1627	1732	4236	2108
	C*0702	2551	5861	2388	4231	4327	1721	2066	4513	2605
	C*0801	1510	3273	1467	2541	2896	1004	1218	3105	1555
	C*1203	1588	3638	1551	2763	3129	1279	1382	3905	1674
	C*1402	2077	5509	2009	3508	3440	1425	1640	3862	2073
	C*1502	1408	3095	1375	2598	2614	1086	1183	3487	1481
	C*1601	1724	4233	1739	3118	3473	1363	1526	3727	1967
	C*1701	1857	4062	2103	3552	3823	1368	1625	3703	1991
	C*1802	2275	4848	2292	3959	3990	1632	2054	4296	2353

The change in the number and strength of sera IgG antibodies against albumin, HLA-A, HLA-B and HLA-C, after affinity purification. Compare with Table 4 which shows the number and strength of these antibodies in raw sera. * Suggesting that the individuals may possibly carry naturally occurring antibodies formed against most commonly shared epitopes.

**Table 6 antibodies-11-00058-t006:** HLA-I isoform reactivity of the monoclonal Abs generated against two alleles of recombinant β2m-free HLA-E heavy chains (Face-2).

Immunogen HLA-E Alleles	Groups	Reactivity of Monoclonal Antibodies with HLA-I Alleles						Number of mAbs
		HLA-Ia			HLA-Ib			
		HLA-A	HLA-B	HLA-C	HLA-E	HLA-F	HLA-G	
HLA-E^R/G107^	A	(–)	(–)	(–)	(+)	(–)	(–)	25
	B	(–)	(–)	(–)	(+)	(+)	(+)	8
	C	A*1101+	(+)	(+)	(+)	(–)	(–)	83
	D	(+)	(+)	(+)	(+)	(–)	(–)	54
	E	(+)	(+)	(+)	(+)	(–)	(+)	18
	F	(+)	(+)	(+)	(+)	(+)	(+)	6
HLA-E^R107^	A1	(–)	B*4006+	(–)	(+)	(–)	(–)	1
	B2	(–)	(+)	(+)	(+)	(–)	(–)	30
	C3	A*1101+	(+)	(+)	(+)	(+)	(–)	1
	D4	A*1101+	(+)	(+)	(+)	(+)	(+)	1
HLA-E^G107^	E5	(–)	(–)	(–)	(+)	(+)	(–)	1
	F6	(–)	(–)	(–)	(+)	(–)	(+)	1
	G7	(–)	(+)	(–)	(+)	(–)	(–)	4
	H8	(+)	(+)	(+)	(+)	(+)	(–)	10

Modified from Ravindranath et al. [104]. * Suggesting that the individuals may possibly carry naturally occurring antibodies formed against most commonly shared epitopes.

**Table 7 antibodies-11-00058-t007:** Recognition pattern of the TFL-006 that binds to exposed cryptic epitopes found in β2m-free HLA HCs (Face-2).

Mean Fluorescent Intensity (MFI)* withTFL-006							
HLA-I	LABSCreen (LS)		LC	HLA-I	LABSCreen (LS)		LC
	Lot # 11	Lot # 10	3005613		Lot # 11	Lot # 10	3005613
	20 mg/mL		20 mg/mL		20 mg/mL		20 mg/mL
A*01:01	852	933	0	B*07:02	670	862	0
A*02:01	542	339	0	B*08:01	1204	1226	0
A*02:03	156	1018	0	**B*13:01**	675	n/a	
**A*02:06**	616	n/a		B*13:02	4923	2514	0
A*03:01	210	193	0	B*14:01	1983	7805	0
A*11:01	3900	4782	0	B*14:02	197	1831	0
A*11:02	321	537	0	B*15:01	2541	335	0
A*23:01	125	133	0	B*15:02	809	1935	0
A*24:02	580	716	0	B*15:03	1146	1822	0
A*24:03	1042	2516	0	**B*15:10**	579	n/a	
A*25:01	349	194	0	**B*15:11**	1658	n/a	
A*26:01	2273	2221	0	B*15:12	1875	770	0
A*29:01	1667	1017	0	B*15:13	2437	3135	0
A*29:02	715	778	0	B*15:16	882	3076	0
A*30:01	1243	1496	0	B*18:01	2891	3096	0
**A*30:02**	728	n/a		B*27:05	3252	634	0
A*31:01	687	396	0	B*27:08	2095	1659	0
A*32:01	341	515	0	B*35:01	517	6128	0
A*33:01	574	1038	0	B*37:01	4352	2650	0
A*33:03	1039	554	0	B*38:01	2425	2521	0
**A*34:01**	3002	n/a		B*39:01	4447	704	0
A*34:02	1165	1535	0	B*40:01	330	3429	0
A*36:01	941	1353	0	B*40:02	3354	2697	0
A*43:01	2768	2479	0	**B*40:06**	1869	n/a	
A*66:01	2145	1886	0	B*41:01	1364	3739	0
A*66:02	1327	1454	0	B*42:01	957	347	0
A*68:01	454	713	0	B*44:02	892	3650	0
A*68:02	578	1185	0	B*44:03	2516	1829	0
A*69:01	1786	3128	0	B*45:01	2568	1736	0
A*74:01	603	652	0	B*46:01	1415	3572	0
A*80:01	3758	3132	0	B*47:01	1859	2152	0
				B*48:01	2418	3262	0
Cw*01:02	2820	4066	0	B*49:01	3042	1554	0
Cw*02:02	3403	7446	0	B*50:01	4246	1799	0
**Cw*03:02**	6614	n/a		B*51:01	2413	2461	0
Cw*03:03	3071	2458	0	**B*51:02**	696	n/a	
Cw*03:04	8167	4504	0	B*52:01	1745	2146	0
Cw*04:01	4182	3337	0	B*53:01	695	5442	0
Cw*05:01	2567	9124	0	B*54:01	2427	1662	0
Cw*06:02	5794	5644	0	B*55:01	4867	2519	0
Cw*07:02	1689	8702	0	B*56:01	3084	3662	0
Cw*08:01	4479	6090	0	B*57:01	141	2089	0
**Cw*12:03**	6731	n/a		**B*57:03**	1878	n/a	
Cw*14:02	3627	3937	0	B*58:01	3190	5268	0
Cw*15:02	4651	4465	0	B*59:01	1278	3553	0
Cw*16:01	2706	4648	0	B*67:01	3520	406	0
Cw*17:01	5760	8296	0	B*73:01	3565	1423	0
**Cw*18:02**	829	n/a	0	B*78:01	5271	2996	0
				B*81:01	2446	1525	0
				**B*82:02**	1959	n/a	

The mAb TFL-006 did not recognize the intact or Face-1 HLA-I molecules coated on the LIFECODES (LC) SABs but recognized LABScreen SAB lots, confirming the presence of Face-2 on the regular LABScreen SABs. Therefore, LABScreen SABs may not be reliable for measurement of the strength of IgG allo-Abs against intact HLA (Face-1). * MFI less than 30 are indicated as 0. Alleles in Bold are antigens unique to LABScreen.

**Table 8 antibodies-11-00058-t008:** Dose-dependent inhibition of blastogenesis and proliferation of activated lymphocytes.

Octagam-IVIg		CD4+ T-Lymphoblasts								CD8+ T-Lymphoblasts							
		Blastogenesis			Proliferation					Blastogenesis			Proliferation				
					M1/2		M3-5						M1/2		M3-5		
		Mean	SD	*p^2^*	Mean	SD	Mean	SD	*p^2^*	Mean	SD	*p^2^*	Mean	SD	Mean	SD	*p^2^*
no PHA		46	8		73	17	1	1		53	13		80	20	4	7	
with PHA only		1685	89	<0.0001	338	15	1486	95	<0.0001	1951	171	<0.0001	329	15	1751	193	<0.0001
PHA + IVIg (1/10)		945	87	0.0005	405	5	675	68	0.0003	1134	13	0.001	431	115	893	9	0.002
PHA + IVIg (1/20)		1365	100	0.019	432	23	1105	71	0.005	1717	198	NS	380	17	1596	46	0.005
PHA + IVIg (1/40)		1796	81	NS	464	31	1540	43	NS	2280	127	NS	419	61	2036	137	NS
TFL-006 (ascites)	Concentration																
No PHA		39	6		53	13	1	1		70	3		53	13	1	1	
With PHA		1977	62	<0.0001	320	3	1850	64	<0.0001	403	31	0.003	320	3	1850	64	<0.0001
(1/7)	127.0 µg/mL	934	69	<0.0001	506	30	576	51	<0.0001	176	75	0.02	506	30	576	51	<0.0001
(1/70)	12.67 µg/mL	1259	105	<0.0001	470	22	954	118	0.0003	191	72	0.02	470	22	954	118	0.0003
(1/140)	6.336 µg/mL	1437	30	0.0002	485	27	1162	57	0.0002	207	67	0.02	485	27	1162	57	0.0002
(1/280)	3.168 µg/mL	1660	110	0.01	404	20	1450	120	0.007	248	16	0.005	404	20	1450	120	0.007
(1/560)	1.584 µg/mL	1851	72	NS.	390	21	1669	75	0.03	477	52	NS	390	21	1669	75	0.03
TFL-007 (ascites)																	
No PHA		34	5		40	6	3	3		34	5		40	6	3	3	
With PHA		2198	224	<0.0001	333	15	2021	221	<0.0001	2198	224	<0.0001	333	15	2021	221	<0.0001
(1/7)	89.57 µg/mL	1050	67	0.001	462	27	714	60	0.0006	1050	67	0.001	462	27	714	60	0.0006
(1/70)	8.957 µg/mL	1372	106	0.0050	473	35	1072	38	0.0020	1372	106	0.0050	473	35	1072	38	0.0020
(1/140)	4.937 µg/mL	1789	132	0.05	521	71	1471	142	0.02	1789	132	0.05	521	71	1471	142	0.02
TFL-037 & TFL-033		Failed to Suppres Blastogenesis or Proliferation															
HLA-I Reactivity of IVIg (Octagam, Mexico)		**HLA-A**			**HLA-B**		**HLA-C**				**HLA-E**		**HLA-F**			**HLA-G**	
		30			47		16				+		+			+	
TFL-006		31			48		16				+		+			+	
TFL-007		26			44		16				+		+			+	
TFL-037		11			37		14				+		0			0	
TFL-033		0			0		0				+		0			0	

Blastogenesis and proliferation (as measured by carboxyfluorescein succinimidyl ester (CFSE)-labelled lymphocytes) of phytohaemagglutinin (PHA)-activated CD4+ and CD8+ T lymphoblasts in vitro by Intravenous Immunoglobulin (IVIg) and HLA-I polyreactive monoclonal Abs (mAbs) TFL-006a and TFL-007a. The data presented is modified from Ravindranath et al. [113,133]. NS: not significant.

## Data Availability

No new data were created or analyzed in this study. Data sharing is not applicable to this article.

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
