# Peer review of "Role of HLA-I Structural Variants and the Polyreactive Antibodies They Generate in Immune Homeostasis"

_2073-4468, 2022, doi:10.3390/antib11030058_

Round 1

Reviewer 1 Report

The manuscript entitled "Role of HLA-I structural variants and the polyreactive antibodies they generate in immune homeostasis" by Ravindranath et al. summarizes new aspects of immune homeostasis from the perspective of Human Leukocyte Antigens and corresponding antibodies against specific regions of the HLA molecules which are only accessible under special pathophysiological conditions.

In this detailed Review the authors cover a wide range from well known HLA basics to own previously published work of the immune homeostatic role of anti-HLA antibodies biding only to epitopes on beta 2 microglobulin free HLA-class I alpha chains. This content is of high scientific interest for the community and partially represents new results and approaches.

The introduction should be shortened. I would recommend to condense Figure 4 as it does not contain information absolutely necessary for the message of this part of the manuscript. Only the very frequent HLA-alleles are shown in Figure 4, and even those are not needed to be depicted as complete amino acid sequences here. The selection of the HLA alleles in Figure 4 is not explained but it can be assumed that only the alleles represented on the Single Antigen Luminex Beads are shown.

Chapter 5 should also be significantly shortened to better represent the character as a review document. The Figures and Tables of this chapter show too many details from previously published work.

Minor points:

Greek letters, e.g. ‘beta’, are often not depicted throughout the manuscript. Some parts are difficult to follow and to understand due to missing letters (e.g. page 4, chapter 2.2). Font sizes differ and should be unified (page 3, chapter 2.1).

Page 12, chapter 3.1: The old HLA nomenclature is not cited correctly (reference 24).

The legends of Table 4, Table 5, Figure 7 and Figure 9 contain incomplete information. MFI values are shown which is only written once in the text body.

Table 6: Please use correct HLA nomenclature.

Author Response

We profusely thank the reviewer for considering that the “content is of high scientific interest for the community and partially represents new results and approaches”. The attached pdf version of our reply will show the steps we have taken to revise the manuscript based on your comments.

Reviewer 2 Report

In the manuscript “Role of HLA-I structural variants and the polyreactive antibodies they generate in immune homeostasis.”, the authors reviewed structural variants of HLA-I and its unique epitopes in detail. Especially, effects of mAb against such epitope to activated lymphocytes are introduced. The review is well-written concerning structure of HLA-I variants. However, description about effects of TFL-006/007 mAbs are somewhat redundant and can be modified to be more concise.

Major comments:

1.       In section 5.2.1, inhibitory effects of IVIg and TEL-006/007 mAbs on activated lymphocytes are reviewed. As for its molecular mechanisms, they describe in Figure 13 but little in the main text. Please describe it in main text with some references.

2.       In section 5.2.2, authors introduced the observation the TFL-006/007 mediated inhibition of antibody secretion from HML16 hybridoma. Although they argue this represents effect of IgIV administration to autoimmune patients, it is unclear if this is limited to anti-allo-HLA antibody production. Rather, it looks like general effect to antibody production from hybridomas.

3.       In section 5.2.3, although IVIg is reported to expand Tregs in mice, only TEF-007 but not IVIgs induced upregulation of Tregs in humans. Is there any publication reporting expansion of Tregs by IVIg in humans?

4.       Sections 6.2.1 and 6.2.2 are quite similar to previous sections. Especially, these sections are referring figures and tables that have been explained in the previous sections.

Minor comments:

1.       P11, line 28, there is no description about mAbs 4A11, 3D11 and 4B4.

2.       Section number 2.4 is duplicated.

3.       Section number 6.2.2 is duplicated.

Author Response

We profusely thank the reviewer for considering that the article “is of high scientific interest for the community and partially represents new results and approaches.”

The attached pdf version of our reply will show the steps we have taken to revise the manuscript based on your comments.

Reviewer 3 Report

The authors present a review entitled “Role of HLA-I structural variants and the polyreactive antibodies they generate in immune homeostasis”. As reported in Conclusion “This review highlights the importance of both polyreactive as well as the monospecific mAbs in understanding the role of HLA-I in immune homeostasis. Confirming the epitope specificity of the mAbs by dosimetric peptide inhibition is a necessary pre-requisite for determining their usefulness in specific immunodiagnosis as well as for developing immunotherapy.”

Overall, the manuscript is well written and after some minor revision could be suitable for Antibodies. My major concerns are the followings:

·      The abstract is too long. It can be reduced by HALF without losing the key points. As reported by guideline the abstract should be a total of about 200 words maximum.

·      references reported in the text are wrong (e.g., Page 25 Tran et al. [103]; Reed and her team [78-91]. Moreover, in the text reference are often reported in round bracket (e.g., page 20). 

·      There are numerous errors of sentences throughout the manuscript (e.g., page 12, lines 28-29).  Some sentences are written in a larger font (e.g., page 11).  Please, carry out a diligent editing of the text. 

·      I recommend avoiding the use of the abbreviation the first time it was used in the text. After you define an abbreviation (regardless of whether it is in parentheses), use only the abbreviation. Do not alternate between spelling out the term and abbreviating it. Please, carry out a diligent editing of the text.

·      There are two paragraphs 6.2.2.

·      graphical abstract is necessary to summarize the main point of the paper.

·      The content of the conclusion Section could be improved to better discuss about the results.

  • Finally, a mention should be done to seminal works about the role of HLA variants in autoimmune diseases onset, consider mentioning PMID: 29395276, PMID: 31777841
  •  

Author Response

We profusely thank the reviewer for considering that the manuscript “is well written and after some minor revision could be suitable for publication.

We have carefully reviewed your major and minor concerns and we have modified the manuscript in response to your concerns, as shown in the attached pdf file
